# THE UNREASONABLE EFFECTIVENESS OF RANDOMIZED REPRESENTATIONS IN ONLINE CONTINUAL GRAPH LEARNING

## ABSTRACT

Catastrophic forgetting is one of the main obstacles for Online Continual Graph Learning (OCGL), where nodes arrive one by one, distribution drifts may occur at any time and offline training on task-specific subgraphs is not feasible. In this work, we explore a surprisingly simple yet highly effective approach for OCGL: we use a fixed, randomly initialized encoder to generate robust and expressive node embeddings by aggregating neighborhood information, training online only a lightweight classifier. By freezing the encoder, we eliminate drifts of the representation parameters, a key source of forgetting, obtaining embeddings that are both expressive and stable. When evaluated across several OCGL benchmarks, despite its simplicity and lack of memory buffer, this approach yields consistent gains over state-of-the-art methods, with surprising improvements of up to 30% and performance often approaching that of the joint offline-training upper bound. These results suggest that in OCGL, catastrophic forgetting can be minimized without complex replay or regularization by embracing architectural simplicity and stability.

## 1 INTRODUCTION

In Online Continual Graph Learning (OCGL), nodes arrive sequentially, are observed only once, and undergo drifts in both distribution and task to solve (Donghi et al., 2025). Therefore, models must adapt to new operating conditions on the fly, while making anytime predictions and retaining past knowledge from limited observations, and under strict memory and latency constraints. This makes OCGL one of the most challenging continual learning scenarios. This setting poses several additional requirements to the traditional Continual Learning (CL) (Parisi et al., 2019; De Lange et al., 2022; Van De Ven et al., 2022), Online CL (Mai et al., 2022), and Continual Graph Learning (CGL) (Zhang et al., 2022a), enabling applications that require fast adaptations and anytime predictions (Koh et al., 2021) such as in healthcare (Le Baher et al., 2023), IoT intrusion detection (Lin et al., 2024) and financial markets (Weber et al., 2019), other than modeling citation networks (Liu et al., 2021; Zhou & Cao, 2021), transportation networks (Chen et al., 2021) and recommender systems (Caroprese et al., 2025).

In this paper, we introduce a simple, yet surprisingly effective approach for node-level OCGL, decoupling node representations from the predictive model, and yielding excellent prediction accuracy while taming the forgetting of previously learned concepts. Our approach leverages the effectiveness of randomized and over-parametrized architectures to provide rich, untrained representations, combined with a lightweight trained classifier on top (Rahimi & Recht, 2008; Rudi & Rosasco, 2017b; Scardapane & Wang, 2017). As demonstrated in related literature, fixed representations from pretrained models help prevent forgetting at the feature extraction level even when used in CL for image classification (Hayes & Kanan, 2020; Pelosin, 2022; Mehta et al., 2023). The beneficial effect appears to be even more striking in graphs, where stable neighborhood embeddings eliminate the need to retain the entire topological information for replay strategies (Zhang et al., 2024a). Additionally, the literature on randomized models provides us results that ensure good approximation and generalization properties for this family of models, given that we use a sufficiently large embedding dimension (Scardapane & Wang, 2017). Coupled with a Streaming Linear Discriminant Analy-

sis (SLDA) classifier (Hayes & Kanan, 2020), this approach generally outperforms state-of-the-art OCGL methods, without need for a memory buffer.

Our main contributions are the following.

1. We propose a surprisingly strong OCGL method. A simple, yet effective method coupling principled untrained features with a lightweight streaming classifier, granting both expressivity and forgetting-resilience.
2. We demonstrate how the approach achieves generally best results across seven OCGL benchmarks, in both class-incremental and time-incremental setups.
3. We theoretically motivate our research findings, opening to the development of new, simpler CL methods without compromising on accuracy.

Beyond providing a strong new method with said advantages, our findings suggest that future OCGL research should rethink model designs, emphasizing architectural simplicity and stability.

## 2 RELATED WORKS

In this section, we first outline the specifics of Online Continual Learning for graphs, the setting used for our experiments, distinguishing it from related paradigms. We then discuss randomized neural models and other fixed feature extraction strategies in Continual Learning. These concepts provide context for the problem addressed in the paper and motivate our design choices.

**Online continual learning for graphs** Continual Learning has been studied mainly in domains such as computer vision (Rebuffi et al., 2017; Lopez-Paz & Ranzato, 2017) or reinforcement learning (Kirkpatrick et al., 2017; Rolnick et al., 2019), with clearly defined task boundaries and no dependencies between successive tasks. In contrast, working on a node level in a graph domain introduces structural dependencies between data points, complicating the definition of the task. Continual Graph Learning (CGL) (Febrinanto et al., 2023; Yuan et al., 2023; Zhang et al., 2024b) extends CL to graph settings, and assumes that the graph is presented in blocks, that is, as subsets of nodes connected in subgraphs (Zhang et al., 2022a). Several methods for CL on graph data have been proposed (Zhou & Cao, 2021; Liu et al., 2021; 2023; Hoang et al., 2023; Sun et al., 2023; Cui et al., 2023), yet, similarly to traditional CL, state-of-the-art performance is achieved by replay-based methods which leverage a memory buffer tailored to leverage graph topology (Zhang et al., 2022b; 2024a). However, the use of Graph Neural Network (GNN) (Scarselli et al., 2009; Micheli, 2009; Kipf & Welling, 2017) models poses an issue regarding task separation: nodes form connections also with nodes that were observed in past tasks, thus requiring access to past information due to message passing (Gilmer et al., 2017). This is sometimes addressed by ignoring inter-task edges (Zhang et al., 2022a), yet it is an unrealistic solution since it assumes supervisory information in the form of task identifiers also at test time. On the other hand, due to small-world nature of most real-world graphs, accessing the entire neighborhood information of the nodes would likely mean using the entirety of past data, rendering the use of the CL approach less meaningful. In this context, a more principled setting is Online Continual Graph Learning (OCGL) (Donghi et al., 2025), which bridges the gap between existing research on online CL and CGL. This setting addresses the issue of access to node neighborhoods with the use of neighborhood sampling, to keep a constant computational footprint even as the graph gets denser over time. In particular, the online setting for CL entails a single pass over the streaming data (Chaudhry et al., 2018; Mai et al., 2022; Soutif–Cormerais et al., 2023), in contrast to traditional offline CL where training is done offline in a batched setting on each task (De Lange et al., 2022). This constraint, together with stricter computational and memory requirements, is motivated by applications where quick model adaptation is necessary, and anytime predictions may be required (Koh et al., 2021), thus impeding the task-wise offline training.

**Randomized representations** In the literature on neural networks, multiple approaches have leveraged some form of randomization, due to training efficiency, and often theoretical guarantees (Scardapane & Wang, 2017). Notable examples for vector data are Random Vector Functional-Link networks (RVFL) (Pao & Takefuji, 1992; Pao et al., 1994), Extreme Learning Machines (ELM) (Huang et al., 2006; 2015) and Echo State Networks (ESN) (Jaeger & Haas, 2004). These fix most network parameters randomly, from an appropriate distribution, training only lightweight readouts. Theoretical results such as universal approximation properties and generalization guarantees have

been proved for some these architectures (Liu et al., 2012). Randomized strategies have also been explored for kernel approximation, such as with Random Fourier Features (RFF) (Rahimi & Recht, 2007), a randomized projection that approximates the RBF kernel. Some works have also explored randomized strategies for graph data, such as with GESN (Gallicchio & Micheli, 2010) or MRGNN (Pasa et al., 2022). Different instantiations of untrained Graph Convolutional Networks (Kipf & Welling, 2017) have been proposed in recent years: GCELM (Zhang et al., 2020) uses a single GCN layer, with randomly initialized and fixed parameters. GCN-RW (Huang et al., 2023) improves on this approach by considering an increased receptive field, employing the square of the adjacency matrix for aggregation. The more recent UGCN (Navarin et al., 2023a) instead uses multiple GCN layers, followed by non-differentiable pooling, as the network does not need to be trained. Inspired by RFF, GRNF (Zambon et al., 2020) are derived from expressive GNN architectures, and constitute a family that can separate graphs.

**Fixed feature extraction for CL**  Some works in the CL literature for image classification suggest the use of a frozen CNN backbone, pre-trained on a different dataset, and then training continually only a classifier such a simple MLP (Van De Ven et al., 2022; Mehta et al., 2023). In the context of graph data, this approach is not currently feasible, as it is not trivial to obtain pre-trained GNNs that can handle graphs with different input feature domains, even though there is ongoing promising work on graph foundation models (Wang et al., 2025). Additionally, despite the many randomized approaches for representation extraction, only a very limited number of CL papers rely on them, and only with image classification tasks. CRNet (Li & Zeng, 2023) uses a randomly initialized network for feature extraction, training the classification head via closed-form solution on each task. RanPAC (McDonnell et al., 2023) uses random projections of feature extracted with a pre-trained model with the objective of increasing dimensionality to facilitate class separation. More recently, RanDumb (Prabhu et al., 2025) leverages Random Fourier Features (Rahimi & Recht, 2007) used with nearest class mean classification. On the other hand, there have been some works that investigated theoretically the use of linear models in simple CL setups (Evron et al., 2022; 2023; Ding et al., 2024), leading us to adopt these simple classifiers. Furthermore, it has been observed that prototype based classifiers, such as nearest class mean, are particularly suited for online CL, due to being schedule-robust, i.e. independent from stream ordering (Wang et al., 2022).

## 3 METHOD

The OCGL setting (Donghi et al., 2025) considers an incremental graph $\mathcal{G}$, induced by a stream of nodes $v_1, v_2, \ldots, v_t, \ldots$ that are added one by one. At each timestep $t$, the graph is updated with neighborhood and attribute information about the incoming node $(v_t, \mathcal{N}(v_t), \boldsymbol{x}^t)$, obtaining an updated graph snapshot $\mathcal{G}^{(t)} = (\mathbb{V}^{(t)}, \mathbb{E}^{(t)}, \boldsymbol{X}^{(t)})$. For example, in the Elliptic dataset (Weber et al., 2019), when a new transaction is processed, its information is captured as node features, and payment flows indicate connections. The goal is to learn a model $F_\Theta$ to make node-level predictions $\hat{y}_v^{(t)}$ for given node $v$ at time step $t$ given information coming from its neighboring nodes in $\mathcal{G}^{(t)}$. Specifically, models are trained with a single pass over the node stream, using information from the associated $l$-hop ego-graph $\mathcal{G}_v^{(t,l)}$. The online setting requests bounded memory and computational cost at each time steps, even as the graph grows. Accordingly, only a subset $\widetilde{\mathcal{G}}_v^{(t,l)}$ of the ego-graph may be allowed to produce prediction

$$\hat{y}_v^{(t)} = F_\Theta \left( \widetilde{\mathcal{G}}_v^{(t,l)} \right) . \tag{1}$$

In the context of CL, the node stream can encompass diverse drift patterns, such as those triggered by a class-incremental setting. Nonetheless, model predictions may be requested for nodes observed in the past, therefore requesting the model not only to adapt to evolving graph conditions but also to preserve previously learned knowledge.

### 3.1 PROPOSED SOLUTION

To address forgetting in the OCGL setting, we propose a decoupled approach to node prediction. In this setup, model $F_\Theta = \Phi \circ \Psi$ decomposes as a node feature extractor $\Psi$ followed by a node-level predictor $\Phi$. Feature extraction is performed by a fixed (randomly initialized) backbone $\Psi$:

$$\boldsymbol{z}_v^{(t)} = \Psi \left( \widetilde{\mathcal{G}}_v^{(t,l)} \right) , \tag{2}$$

where $z_v^{(t)} \in \mathbb{R}^d$ denotes the node embedding that captures neighborhood information. As $\Psi$ is untrained, it does not change over time and is inherently immune to forgetting. Secondly, it offers a clear advantage for experience replay methods, as storing $z_v^{(t)}$ is significantly more memory-efficient than storing the entire sampled neighborhood $\widetilde{\mathcal{G}}_v^{(t,l)}$.

Feature extractor $\Psi$ also mitigates other sources of forgetting. Unlike trained models, which tend to produce task-specific representations that remove task-irrelevant information, a well-chosen $\Psi$ can produce rich node embeddings suitable for multiple tasks. Notably, theoretical results (Rudi & Rosasco, 2017a) show that for some families of model architectures, a random initialization of $\Psi$ yields linearly separable embeddings, allowing to effectively solve the downstream task by simple linear predictive models

$$\hat{y}_v^{(t)} = \Phi_{\boldsymbol{W},\boldsymbol{b}}\left(z_v^{(t)}\right) = \boldsymbol{W} z_v^{(t)} + \boldsymbol{b}, \tag{3}$$

which are less prone to forgetting than deep networks (Mirzadeh et al., 2022) and can be learned efficiently.

## 3.2 RANDOMIZED FEATURE EXTRACTION

We consider two different types of graph feature extractors $\Psi$, or backbones, for our experiments, untrained GCN (Navarin et al., 2023a) and Graph Random Neural Features (Zambon et al., 2020) adapted to the node-level, which we discuss here. In general, any randomized graph network with sufficient expressivity and robustness to limited structural shifts can be adopted for our decoupled approach.

**UGCN**  As first feature extractor, wee use the recent UGCN (Navarin et al., 2023a), which uses multiple GCN layers $\boldsymbol{H}^{(i)} = tanh(\tilde{\boldsymbol{A}} \boldsymbol{H}^{(i-1)}\Theta)$, where $\tilde{\boldsymbol{A}}$ is the normalized adjacency matrix, with $\boldsymbol{H}^{(0)} = \boldsymbol{X}$ and with weights initialized with a Glorot uniform approach (Glorot & Bengio, 2010) regulated by a gain hyperparameter, and successively left untrained. In our case, since we adopt it for node-level predictions instead of graph-level ones, we do not use the entire adjacency matrix, but the efficient forward propagation limited to the node ego-graph of (Hamilton et al., 2017). The output of the different layers is then concatenated to obtain an embedding that reflects information at multiple resolutions, i.e.

$$\Psi_{\text{UGCN}}\left(\widetilde{\mathcal{G}}_v^{(t,l)}\right) = \text{concat}\left(\boldsymbol{H}_v^{(1)}, \ldots, \boldsymbol{H}_v^{(l)}\right). \tag{4}$$

In an over-parameterized regime, this model performs competitively with trained GCN counterparts on graph classification (Navarin et al., 2023b; Donghi et al., 2024).

**GRNF**  Graph Random Neural Features (Zambon et al., 2020) define expressive embeddings for attributed graphs, derived from a GNN model which is a universal approximator of graph functions (Maron et al., 2019; Keriven & Peyré, 2019). Each GRNF is a parametric maps $\psi(\cdot, \boldsymbol{w}) : \mathcal{T}^2 \to \mathbb{R}$ defined by the composition of equivariant and invariant affine maps, interleaved with non-linearity:

$$\psi_\omega(\cdot) = \sigma \circ H_k(\cdot, \theta_H) \circ \sigma \circ F_{2,k}(\cdot, \theta_F), \tag{5}$$

where $\omega = (k, \theta_F, \theta_H)$, and $k$ represent tensor order (formal definitions of $F_{2,k}$ and $H_k$, and more details can be found in (Zambon et al., 2020)). Interestingly, the family of GRNF can separate graphs, meaning that for any non-isomorphic graphs $\mathcal{G}_1, \mathcal{G}_2$, there exists $\omega$ such that $\psi_\omega(\mathcal{G}_1) \neq \psi_\omega(\mathcal{G}_2)$. Thus, under suitable assumptions on the distribution of $\boldsymbol{w}$, the family of GRNF induces a metric distance on graphs, with also results that hold in probability for a fixed number of features (Zambon et al., 2020). To adapt GRNF to make node level predictions, we use them on $\widetilde{\mathcal{G}}_v^{(t,l)}$, and since the the output is permutation invariant over the entire ego-graph, we concatenate it to the component relative to $v$ of the output of the equivariant map, using $d/2$ independent draws of $\omega$, resulting in a $d$-dimensional embedding (assuming even $d$).

## 3.3 LINEAR CLASSIFIER

As indicated in equation 3, the extracted features are fed to a linear layer, which can be trained continually on the stream. Effectively, thanks to this strategy we have simplified the Continual Graph Learning problem by eliminating the graph specificity, relegating it to the fixed feature extractor,

and reducing the problem to Continual Learning on vectors. A linear layer may thus be trained with gradient descent, supported by any standard CL learning strategy, such as experience replay which can now leverage informative node embeddings.

Furthermore, we can also address forgetting in the linear classifier by adopting a specific form which does not require gradients: we use Streaming Linear Discriminant Analysis (SLDA) (Hayes & Kanan, 2020), which learns by accumulating class means, independently for each class. This choice makes training more efficient and provides further robustness against catastrophic forgetting risk, as further discussed in Section 3.4. Specifically, during training, SLDA keeps a cumulative mean $\mu_y$ for the features of each class $y$, and a shared covariance matrix $\Sigma$ with streaming updates. To make predictions, using precision matrix $\Lambda = [(1 - \epsilon)\Sigma + \epsilon \boldsymbol{I}]^{-1}$ with $\epsilon = 10^{-4}$, the parameters $\boldsymbol{w}_y$ (rows of $\boldsymbol{W}$) and $\boldsymbol{b}$ of equation 3 are computed as

$$\boldsymbol{w}_y = \Lambda \mu_y \,, \qquad \boldsymbol{b}_y = -\frac{1}{2}(\mu_y^T \Lambda \mu_y) \,. \qquad (6)$$

### 3.4 EFFECTIVENESS OF THE APPROACH

As the graph evolves, multiple sources of forgetting can impair the model's prediction accuracy. This section elaborates on our method's effectiveness in mitigating forgetting while yielding high anytime classification accuracy.

One is a common challenge in all CL setups and stems from the continual update of the model's parameters based on recent training signals, both for the feature extraction backbone (if trained) and for the classifier. A second, graph-specific issue arises from the evolving nature of the graphs and associated structural shifts (Su et al., 2023; 2024): predictions made for the same node at different time steps may rely on different and potentially inconsistent neighborhoods. To put our intuitions more formally, we consider time interval $\delta t$ and define the forgetting risk for node $v$ at time $t$ as the increase in the loss from time $t_0 = t - \delta t$ to time $t$:

$$\Delta \mathcal{R}_v = \Delta \mathcal{R}_{v,t}(\delta t) = \ell\left(y, \Phi_{W_t}\left(\Psi_{\Theta_t}\left(\mathcal{G}_v^t\right)\right)\right) - \ell\left(y, \Phi_{W_{t_0}}\left(\Psi_{\Theta_{t_0}}\left(\mathcal{G}_v^{t_0}\right)\right)\right) \,, \qquad (7)$$

with loss function $\ell$ and where we made explicit the decomposition into feature extraction ($\Psi_\Theta$) and linear classifier ($\Phi_{\mathbf{W}}$); to avoid overwhelming notations, we omit the bias terms in $\Phi_{\mathbf{W}}$ and use $\mathcal{G}_v^t$ to indicate $\widetilde{\mathcal{G}}_v^{(t,l)}$. Then, assuming that the loss is Lipschitz continuous with constant $L_\ell$, and using the triangle inequality, we can isolate three components of forgetting risk:

$$\Delta \mathcal{R}_v \leq L_\ell \left( \left\| \Phi_{W_t}\left(\Psi_{\Theta_t}\left(\mathcal{G}_v^t\right)\right) - \Phi_{W_t}\left(\Psi_{\Theta_t}\left(\mathcal{G}_v^{t_0}\right)\right) \right\| \right. \qquad \textit{structural drift} \qquad (8)$$

$$+ \left\| \Phi_{W_t}\left(\Psi_{\Theta_t}\left(\mathcal{G}_v^{t_0}\right)\right) - \Phi_{W_t}\left(\Psi_{\Theta_{t_0}}\left(\mathcal{G}_v^{t_0}\right)\right) \right\| \qquad \textit{backbone parameters drift} \qquad (9)$$

$$\left. + \left\| \Phi_{W_t}\left(\Psi_{\Theta_{t_0}}\left(\mathcal{G}_v^{t_0}\right)\right) - \Phi_{W_{t_0}}\left(\Psi_{\Theta_{t_0}}\left(\mathcal{G}_v^{t_0}\right)\right) \right\| \right) \,. \qquad \textit{classifier parameters drift} \qquad (10)$$

While the first term is inherent in the data-generating process, and therefore irreducible, a crucial advantage of using a fixed feature extractor is that it eliminates the second term, as $\Theta_t = \Theta_{t_0}$. Furthermore, if we assume the norm of the embeddings to be bounded by $B_z$, the third term is bounded by $B_z \|\boldsymbol{W}_t - \boldsymbol{W}_{t_0}\|$. For SLDA, the term is expected to decrease as the number of observed examples of a class increases, due to the update scheme through separate, class-specific cumulative means; this is in contrast with SGD-based training, where weight updates are less separated by class. Therefore, SLDA with fixed feature extraction provides high stability, with decreasing forgetting risk as the node stream progresses. Higher stability compared to SGD-trained methods can be empirically observed in Figures 2-3 of Appendix F.

This analysis illustrates the robustness against forgetting of untrained feature extraction with SLDA. However, stability is not sufficient for good CL performance: the model must also remain plastic enough to acquire new knowledge as the node stream evolves. For this, we can rely on the over-parameterized randomized feature extractors, which give us expressive and general topological embedding independently of task, allowing for the training of just a simple linear classifier on top (Scardapane & Wang, 2017).

## 4 EXPERIMENTAL SETTING

For our experiments we adopt the OCGL setting described in (Donghi et al., 2025). In particular, we follow the requirement of neighborhood sampling, and we consider small mini-batches of nodes

instead of individual ones. We compare the use of randomized feature extractors, UGCN and GRNF, with linear classifier, either SLDA or coupled with CL strategies, across multiple benchmarks.

**Benchmarks**   We use the same six node-classification graph datasets of (Donghi et al., 2025): CoraFull (Bojchevski & Günnemann, 2018), Arxiv (Hu et al., 2021), Reddit (Hamilton et al., 2017), Amazon Computer (Shchur et al., 2019), Roman Empire (Platonov et al., 2022) and Elliptic (Weber et al., 2019). On all except Elliptic (as it only has two classes) we consider a class-incremental stream: nodes in the graph arrive one by one in blocks consisting of two classes (each segment can be identified with a task in the context of CL, even though the models in this setting are agnostic to task boundaries). On Elliptic and Arxiv we consider a time-incremental stream: since real node timestamps are available, we use them for a realistic node stream (dividing the stream into 10 blocks simply for evaluation). We split the nodes in each graph into 60% for training, 20% for validation and 20% for testing, and we use a transductive setting. We consider the same mini-batch sizes as (Donghi et al., 2025): 10 for the smaller CoraFull, Amazon Computer and Roman Empire, 50 for Arxiv, Reddit and Elliptic.

**Metrics**   To evaluate model predictions in the considered setting, we use three metrics: *Average Performance (AP)*, *Average Forgetting (AF)* (Lopez-Paz & Ranzato, 2017), and *Average Anytime Performance (AAP)* (Caccia et al., 2021). The performance metric is accuracy for all datasets except for Elliptic, as it is highly unbalanced with only two classes, and therefore F1 score of the minority class is used. For anytime predictions, we obtain AAP by evaluating the model on the validation nodes after each training mini-batch. The metrics are described in detail in Appendix B.

**Baselines**   In addition to the described SLDA, we couple the linear classifier with some popular CL strategies: we consider A-GEM (Chaudhry et al., 2018), ER (Chaudhry et al., 2019), EWC (Kirkpatrick et al., 2017), LwF (Li & Hoiem, 2018) and MAS (Aljundi et al., 2018). Furthermore, we consider the *bare* baseline which consists of simply finetuning the linear layer on the stream without any CL method. We also provide the *joint* baseline consisting of jointly training the linear layer offline on the embeddings from all the nodes in the entire final graph. We do not consider graph-specific methods, as once the features are extracted with the untrained backbone they are no longer relevant. However, we provide a comparison with recent state-of-the-art graph methods for OCGL (Donghi et al., 2025) in Appendix E and in summary in Table 3.

**Implementation details**   Following (Donghi et al., 2025), we consider sparsified 2-hop node neighborhoods, sampling recursively 10 neighbors per layer for each node. For UGCN, we therefore use a 2-layer network with 1024 units per layer, resulting in a 2048-dimensional node embedding due to layer concatenation. For GRNF, we use 1024 features, which amount to a 2048-dimensional node embedding since we concatenate the equivariant and invariant components. In both cases, magnitude of weight initialization is regulated by a tunable gain hyperparameter. We use Adam optimizer (Kingma & Ba, 2017) without weight decay nor dropout, tuning the learning rate as an hyperparameter. Another hyperparameter is the number of passes on a mini-batch before passing to the next, as suggested by Aljundi et al. (Aljundi et al., 2019). Hyperparameters are tuned following the protocol outlined by Chaudhry et al. (Chaudhry et al., 2018): they are selected according to validation performance (AP) only on a small section of the data stream. All training and method specific hyperparameters are reported in Appendix D. All experiments are performed with 5 different random initializations, and results are reported as average and standard deviation over them.

## 5   RESULTS

The results of our experiments in the OCGL setting are reported in Table 1 for benchmarks with class-incremental node stream and Table 2 for those on which a time-incremental stream is defined. Additionally, for comparison with results obtainable by a model trained end-to-end, we report in Table 3 the best AP reported in (Donghi et al., 2025) (Arxiv is reported only with class-incremental stream as the time-incremental one is not considered in (Donghi et al., 2025)) by any OCGL strategy, in most cases SSM (Zhang et al., 2022b) and PDGNN (Zhang et al., 2024a), together with the respective joint upper bound, and we highlight the increase in performance using the proposed decoupled approach of randomized feature extraction with SLDA.

| | METHOD | UGCN | | | GRNF | | |
|---|---|---|---|---|---|---|---|
| | | AP% ↑ | AAP$_{val}$% ↑ | AF% ↑ | AP% ↑ | AAP$_{val}$% ↑ | AF% ↑ |
| CORAFULL | A-GEM | $32.99_{\pm1.02}$ | $52.35_{\pm0.90}$ | $-50.53_{\pm1.19}$ | $33.33_{\pm0.42}$ | $48.04_{\pm0.89}$ | $-29.93_{\pm0.67}$ |
| | ER | $55.67_{\pm0.47}$ | $65.01_{\pm0.60}$ | $-24.90_{\pm0.71}$ | $52.53_{\pm0.60}$ | $59.79_{\pm0.56}$ | $-27.40_{\pm0.70}$ |
| | EWC | $34.46_{\pm1.28}$ | $52.28_{\pm0.35}$ | $-49.31_{\pm1.82}$ | $30.53_{\pm0.52}$ | $46.66_{\pm0.92}$ | $-50.44_{\pm0.75}$ |
| | LwF | $35.30_{\pm0.50}$ | $52.94_{\pm1.18}$ | $-46.41_{\pm0.74}$ | $30.70_{\pm0.85}$ | $45.87_{\pm0.98}$ | $-35.06_{\pm0.50}$ |
| | MAS | $34.44_{\pm1.30}$ | $52.59_{\pm0.36}$ | $-49.44_{\pm1.79}$ | $30.12_{\pm0.63}$ | $46.58_{\pm0.87}$ | $-51.28_{\pm0.73}$ |
| | SLDA | $64.03_{\pm0.50}$ | $74.65_{\pm0.44}$ | $-14.48_{\pm0.39}$ | $62.05_{\pm0.27}$ | $72.42_{\pm0.31}$ | $-16.87_{\pm0.50}$ |
| | BARE | $32.10_{\pm1.50}$ | $51.48_{\pm0.55}$ | $-51.85_{\pm1.62}$ | $28.18_{\pm0.71}$ | $44.82_{\pm1.11}$ | $-35.56_{\pm0.65}$ |
| | JOINT | $66.10_{\pm0.28}$ | - | $-8.91_{\pm0.36}$ | $65.23_{\pm0.78}$ | - | $-8.05_{\pm0.54}$ |
| A. COMPUTER | A-GEM | $60.35_{\pm4.86}$ | $58.12_{\pm3.18}$ | $-31.60_{\pm7.88}$ | $53.69_{\pm5.85}$ | $62.51_{\pm3.59}$ | $-39.10_{\pm8.55}$ |
| | ER | $80.71_{\pm2.83}$ | $83.30_{\pm0.51}$ | $-13.32_{\pm3.99}$ | $62.62_{\pm10.93}$ | $75.25_{\pm2.93}$ | $-20.04_{\pm11.26}$ |
| | EWC | $48.02_{\pm1.30}$ | $59.01_{\pm1.07}$ | $-30.29_{\pm1.55}$ | $30.80_{\pm5.50}$ | $49.10_{\pm1.05}$ | $-44.21_{\pm2.34}$ |
| | LwF | $47.31_{\pm11.89}$ | $60.84_{\pm3.24}$ | $-30.40_{\pm13.68}$ | $40.59_{\pm5.37}$ | $53.53_{\pm1.97}$ | $-50.96_{\pm2.41}$ |
| | MAS | $41.81_{\pm2.06}$ | $62.90_{\pm0.87}$ | $-45.72_{\pm3.10}$ | $43.14_{\pm3.89}$ | $57.19_{\pm2.01}$ | $-44.81_{\pm3.95}$ |
| | SLDA | $86.65_{\pm0.48}$ | $90.64_{\pm0.13}$ | $-7.72_{\pm0.47}$ | $84.32_{\pm0.42}$ | $89.95_{\pm0.23}$ | $-10.94_{\pm0.39}$ |
| | BARE | $43.98_{\pm3.39}$ | $50.21_{\pm1.26}$ | $-42.56_{\pm8.59}$ | $42.53_{\pm6.22}$ | $52.62_{\pm1.53}$ | $-49.47_{\pm2.49}$ |
| | JOINT | $86.84_{\pm0.47}$ | - | $-7.30_{\pm0.47}$ | $87.43_{\pm0.34}$ | - | $-6.80_{\pm0.33}$ |
| ARXIV | A-GEM | $22.68_{\pm3.24}$ | $30.81_{\pm2.18}$ | $-59.57_{\pm1.86}$ | $14.32_{\pm1.82}$ | $32.39_{\pm1.80}$ | $-64.85_{\pm1.27}$ |
| | ER | $15.12_{\pm4.06}$ | $32.36_{\pm3.98}$ | $-49.55_{\pm4.96}$ | $14.63_{\pm1.78}$ | $31.84_{\pm2.05}$ | $-57.71_{\pm1.41}$ |
| | EWC | $17.26_{\pm2.12}$ | $27.71_{\pm1.02}$ | $-56.43_{\pm2.11}$ | $17.31_{\pm1.06}$ | $26.33_{\pm0.66}$ | $-67.39_{\pm0.52}$ |
| | LwF | $19.52_{\pm1.49}$ | $27.04_{\pm0.88}$ | $-56.52_{\pm0.92}$ | $21.72_{\pm0.98}$ | $29.84_{\pm0.40}$ | $-60.04_{\pm1.71}$ |
| | MAS | $18.77_{\pm2.44}$ | $28.92_{\pm1.27}$ | $-53.20_{\pm3.47}$ | $16.45_{\pm1.35}$ | $29.10_{\pm0.82}$ | $-67.50_{\pm1.20}$ |
| | SLDA | $55.71_{\pm0.08}$ | $64.55_{\pm0.03}$ | $-18.47_{\pm0.08}$ | $52.69_{\pm0.19}$ | $62.69_{\pm0.07}$ | $-27.67_{\pm0.16}$ |
| | BARE | $21.38_{\pm3.37}$ | $24.28_{\pm1.69}$ | $-41.42_{\pm2.17}$ | $14.13_{\pm1.14}$ | $24.67_{\pm0.33}$ | $-75.06_{\pm1.31}$ |
| | JOINT | $59.06_{\pm0.15}$ | - | $-16.09_{\pm0.30}$ | $57.87_{\pm0.28}$ | - | $-16.72_{\pm0.19}$ |
| REDDIT | A-GEM | $60.60_{\pm0.88}$ | $78.29_{\pm0.31}$ | $-36.77_{\pm0.89}$ | $46.51_{\pm0.91}$ | $61.06_{\pm0.20}$ | $-25.63_{\pm0.96}$ |
| | ER | $80.40_{\pm1.04}$ | $89.75_{\pm0.05}$ | $-16.60_{\pm1.11}$ | $83.46_{\pm0.43}$ | $89.22_{\pm0.12}$ | $-14.44_{\pm0.40}$ |
| | EWC | $40.91_{\pm1.22}$ | $61.13_{\pm1.07}$ | $-49.58_{\pm1.37}$ | $44.26_{\pm1.12}$ | $60.49_{\pm0.45}$ | $-29.80_{\pm1.18}$ |
| | LwF | $13.60_{\pm0.43}$ | $49.00_{\pm0.85}$ | $-84.37_{\pm0.43}$ | $39.09_{\pm0.77}$ | $58.00_{\pm1.07}$ | $-35.99_{\pm0.87}$ |
| | MAS | $11.93_{\pm0.66}$ | $50.54_{\pm0.48}$ | $-86.50_{\pm0.66}$ | $42.99_{\pm0.99}$ | $58.22_{\pm0.50}$ | $-32.62_{\pm0.97}$ |
| | SLDA | $89.31_{\pm0.03}$ | $95.17_{\pm0.03}$ | $-5.56_{\pm0.09}$ | $91.42_{\pm0.14}$ | $95.84_{\pm0.03}$ | $-3.81_{\pm0.15}$ |
| | BARE | $41.35_{\pm1.27}$ | $60.55_{\pm1.05}$ | $-48.97_{\pm1.26}$ | $42.67_{\pm0.75}$ | $57.80_{\pm0.36}$ | $-33.13_{\pm0.41}$ |
| | JOINT | $88.07_{\pm0.20}$ | - | $-3.57_{\pm0.19}$ | $90.90_{\pm0.17}$ | - | $-2.77_{\pm0.17}$ |
| ROMAN E. | A-GEM | $17.18_{\pm0.46}$ | $43.99_{\pm0.20}$ | $-69.65_{\pm0.42}$ | $31.19_{\pm0.98}$ | $37.77_{\pm1.42}$ | $-23.28_{\pm4.54}$ |
| | ER | $24.80_{\pm1.21}$ | $46.35_{\pm0.50}$ | $-32.56_{\pm0.84}$ | $40.45_{\pm1.18}$ | $46.83_{\pm1.46}$ | $-15.68_{\pm2.20}$ |
| | EWC | $15.82_{\pm1.00}$ | $37.30_{\pm0.50}$ | $-43.62_{\pm0.95}$ | $21.54_{\pm0.85}$ | $40.22_{\pm0.91}$ | $-22.71_{\pm2.53}$ |
| | LwF | $21.51_{\pm0.65}$ | $45.51_{\pm0.37}$ | $-42.38_{\pm0.83}$ | $20.29_{\pm1.84}$ | $37.25_{\pm0.76}$ | $-46.63_{\pm2.02}$ |
| | MAS | $16.33_{\pm0.66}$ | $35.12_{\pm0.81}$ | $-28.71_{\pm1.77}$ | $22.47_{\pm0.58}$ | $40.52_{\pm0.42}$ | $-26.38_{\pm0.60}$ |
| | SLDA | $34.61_{\pm0.02}$ | $57.57_{\pm0.06}$ | $-34.38_{\pm0.20}$ | $52.71_{\pm0.09}$ | $72.07_{\pm0.03}$ | $-29.43_{\pm0.21}$ |
| | BARE | $7.09_{\pm0.03}$ | $34.74_{\pm0.69}$ | $-77.07_{\pm0.93}$ | $16.82_{\pm0.29}$ | $42.44_{\pm0.66}$ | $-68.56_{\pm1.54}$ |
| | JOINT | $49.12_{\pm0.51}$ | - | $-5.22_{\pm0.48}$ | $71.50_{\pm0.10}$ | - | $-6.54_{\pm0.37}$ |

Table 1: Results for class-incremental node stream.

**Class-incremental benchmarks** We first observe from Table 1 that the upper bounds with randomized features are comparable to those obtained with a trained GCN model, as reported in Table 3. This proves that the extracted features are expressive enough for these tasks, confirming our approach's viability. On Roman Empire specifically, which is highly heterophilic, we see a much higher upper bound with GRNF: this is due to the implicit bias of the GCN design, which smooths node embeddings with neighborhood information, while GRNF are more expressive, as they are derived from a universal approximator of graph functions. We also note how AF for the joint baseline represents the increasing difficulty of the classification task as new classes are added.

From the results the superior performance of SLDA also emerges clearly, as it outperforms all considered CL methods on a linear layer trained with gradient descent, and in most cases with a wide margin. Additionally, the AP results of SLDA with both UGCN and GRNF closely approach their respective upper bounds. The results of the randomized feature extractors coupled with SLDA also significantly outperform the best results obtained with state-of-the-art replay methods tailored for graphs, as seen in Table 3, despite SLDA not using any memory buffer. The differences between UGCN and GRNF are in most case limited, especially for the SLDA classifier, with UGCN showing

| | METHOD | UGCN | | | GRNF | | |
|---|---|---|---|---|---|---|---|
| | | AP% ↑ | AAP$_{val}$% ↑ | AF% ↑ | AP% ↑ | AAP$_{val}$% ↑ | AF% ↑ |
| ELLIPTIC | A-GEM | $42.43_{\pm 0.99}$ | $40.72_{\pm 0.57}$ | $-13.51_{\pm 1.54}$ | $53.65_{\pm 0.73}$ | $50.10_{\pm 0.33}$ | $-12.08_{\pm 1.16}$ |
| | ER | $44.61_{\pm 1.29}$ | $45.54_{\pm 0.40}$ | $-9.17_{\pm 1.82}$ | $57.30_{\pm 1.69}$ | $53.36_{\pm 1.29}$ | $-8.90_{\pm 1.65}$ |
| | EWC | $36.62_{\pm 1.23}$ | $34.64_{\pm 0.20}$ | $-13.36_{\pm 1.59}$ | $50.44_{\pm 1.45}$ | $48.81_{\pm 0.36}$ | $-13.77_{\pm 1.16}$ |
| | LwF | $37.88_{\pm 1.52}$ | $34.63_{\pm 0.30}$ | $-11.43_{\pm 1.39}$ | $55.16_{\pm 1.26}$ | $50.15_{\pm 0.59}$ | $-10.45_{\pm 0.88}$ |
| | MAS | $36.80_{\pm 0.84}$ | $34.47_{\pm 0.21}$ | $-13.21_{\pm 1.30}$ | $50.26_{\pm 1.19}$ | $48.30_{\pm 0.36}$ | $-14.29_{\pm 0.97}$ |
| | SLDA | $55.49_{\pm 0.64}$ | $54.13_{\pm 0.28}$ | $-1.85_{\pm 1.17}$ | $65.92_{\pm 0.71}$ | $65.43_{\pm 0.53}$ | $-3.21_{\pm 1.90}$ |
| | BARE | $37.33_{\pm 0.68}$ | $34.97_{\pm 0.32}$ | $-12.21_{\pm 1.01}$ | $50.14_{\pm 1.44}$ | $48.81_{\pm 0.45}$ | $-13.23_{\pm 1.11}$ |
| | JOINT | $59.29_{\pm 0.82}$ | - | $-4.94_{\pm 0.62}$ | $70.23_{\pm 0.76}$ | - | $-3.59_{\pm 1.00}$ |
| ARXIV | A-GEM | $67.66_{\pm 0.12}$ | $64.97_{\pm 0.07}$ | $0.98_{\pm 0.17}$ | $67.73_{\pm 0.06}$ | $63.07_{\pm 0.11}$ | $2.32_{\pm 0.09}$ |
| | ER | $68.67_{\pm 0.18}$ | $65.26_{\pm 0.05}$ | $2.12_{\pm 0.18}$ | $68.30_{\pm 0.04}$ | $63.56_{\pm 0.09}$ | $2.93_{\pm 0.09}$ |
| | EWC | $67.65_{\pm 0.07}$ | $64.96_{\pm 0.03}$ | $0.94_{\pm 0.08}$ | $67.65_{\pm 0.08}$ | $63.93_{\pm 0.12}$ | $1.48_{\pm 0.10}$ |
| | LwF | $68.21_{\pm 0.09}$ | $65.22_{\pm 0.08}$ | $1.37_{\pm 0.09}$ | $67.56_{\pm 0.12}$ | $62.94_{\pm 0.12}$ | $2.22_{\pm 0.11}$ |
| | MAS | $67.65_{\pm 0.07}$ | $64.96_{\pm 0.03}$ | $0.94_{\pm 0.08}$ | $67.64_{\pm 0.09}$ | $63.93_{\pm 0.12}$ | $1.47_{\pm 0.10}$ |
| | SLDA | $66.69_{\pm 0.07}$ | $61.97_{\pm 0.04}$ | $2.29_{\pm 0.17}$ | $65.09_{\pm 0.21}$ | $60.14_{\pm 0.10}$ | $2.42_{\pm 0.20}$ |
| | BARE | $67.59_{\pm 0.20}$ | $64.98_{\pm 0.04}$ | $0.97_{\pm 0.18}$ | $67.68_{\pm 0.09}$ | $63.89_{\pm 0.04}$ | $1.57_{\pm 0.10}$ |
| | JOINT | $70.35_{\pm 0.16}$ | - | $2.67_{\pm 0.30}$ | $69.53_{\pm 0.12}$ | - | $2.89_{\pm 0.09}$ |

Table 2: Results for time-incremental node stream.

| METHOD | CORAFULL | A. COMPUTER | ARXIV (CL.-INCR.) | REDDIT | ROMAN E. | ELLIPTIC |
|---|---|---|---|---|---|---|
| BEST OCGL | $40.45_{\pm 0.77}$ | $70.45_{\pm 3.66}$ | $35.86_{\pm 1.20}$ | $58.08_{\pm 8.04}$ | $14.20_{\pm 0.87}$ | $51.13_{\pm 1.74}$ |
| JOINT | $67.55_{\pm 0.05}$ | $83.07_{\pm 1.30}$ | $58.58_{\pm 0.28}$ | $90.02_{\pm 0.12}$ | $39.47_{\pm 0.33}$ | $71.97_{\pm 0.83}$ |
| UGCN+SLDA | +23.6 | +16.2 | +19.9 | +31.2 | +20.4 | +4.4 |
| GRNF+SLDA | +21.6 | +13.9 | +16.8 | +33.3 | +38.5 | +14.8 |

Table 3: Top rows: best AP results from Table 5 of Appendix E for CL strategies with trained GCN and joint offline training upper bound. Bottom rows: increase in AP of the proposed approach (Tables 1-2) compared to the best performing method with trained GCN. All the increases in performance show statistical significance with p-value $\leq 0.005$ with a one-sided Mann–Whitney U test.

a slight advantage on CoraFull, Amazon Computer and Arxiv benchmarks, while GRNF appears superior on Reddit and more significantly on Roman Empire, due to the heterophily of the graph as discussed above.

In general, compared to the full results in Appendix E, also most other CL strategies in most benchmarks report performance improvements compared to the use of a trained GCN, confirming the benefits of keeping a frozen feature extractor immune to forgetting. ER specifically is significantly better when coupled with randomized features compared to a trained GCN, in some cases approaching SLDA, due to the fact that in this setting the memory buffer is much more informative, as stored examples contain neighborhood information, albeit subject to structural shift.

**Time-incremental benchmarks** The time-incremental setting naturally possesses more docile shifts in distribution, compared to the abrupt class changes of the class-incremental benchmark. Nonetheless, on Elliptic CL methods are still beneficial, with SLDA still outperforming all baselines, also compared to the trained GCN state-of-the-art (Table 3). Importantly, our results on Elliptic, which is a dataset of real Bitcoin transactions, highlight the feasibility and benefits of the proposed approach on realistic data streams, beyond the academic class-incremental setting. On the other hand, for the Arxiv time-incremental benchmark, we see little difference in the results of the various strategies, with SLDA no longer over-performing. In fact, we even see positive AF values for all methods, indicating that the classification becomes easier, rather than harder, as the node stream goes on. This is because Arxiv does not present a significant drift in class distribution throughout time. Therefore, a CL learning approach here is less meaningful, as the bare baseline is already close to the upper bound. Nevertheless, ER proves beneficial, and these high results are further proof that the feature extractors are robust to structural shifts that still remain.

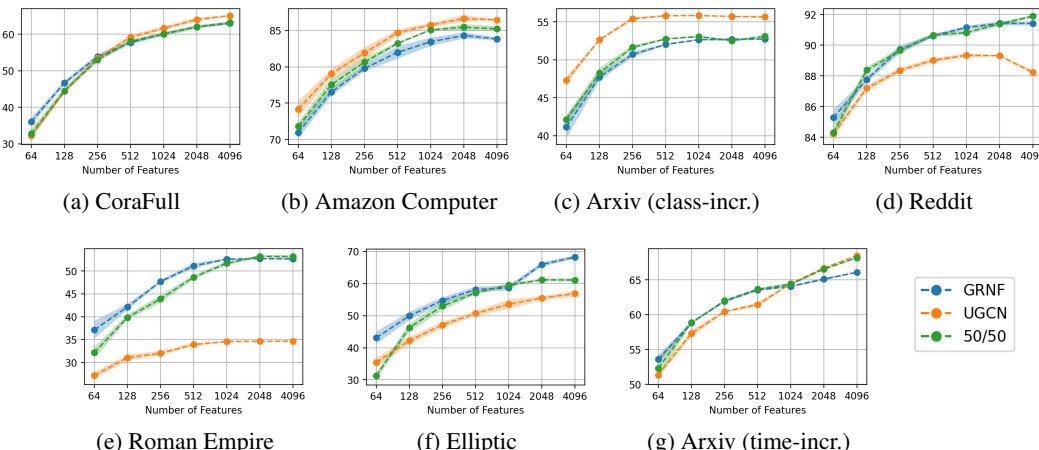

Figure 1: Comparison of AP of SLDA with different number of features extracted with GRNF, UGCN, and a 50/50 mix of the two. The shaded area covers one standard deviation.

**Impact of number of features** Given the overwhelming over-performance of randomized feature extraction with SLDA, we investigate the impact of the number of extracted features on model performance. In Figure 1, we see that, with a lower number of features AP decreases as well, even though on most benchmarks even as few as 64 randomized features are sufficient to obtain results on par with the state-of-the-art CL methods on trained GCN of Table 3. Also, for many benchmarks performance seems to not have reached saturation even at 4096 features, indicating that further gains could be achieved with a larger feature extractor.

Finally, since UGCN and GRNF appear to have different strengths over the multiple benchmarks, we consider a hybrid feature extractor that extracts half of the features with UGCN, and half with GRNF. This mixed feature extractor shows a generally more robust performance over the benchmarks, with performance that is never lower than the individual two, except for one single point. Generally the hybrid extractor also does not improve over the best single one, indicating that the two types of feature do not benefit from integration. However, this strategy could be used to obtain a reliable feature extractor without the need to evaluate the two strategies.

## 6    CONCLUSION

This work introduces a simple, yet surprisingly effective approach for Online Continual Graph Learning, addressing forgetting by decoupling representation learning from classification. We use randomized, fixed node feature extractors that encode neighborhood information, coupled with a lightweight linear classifier trained incrementally on the node stream. By leveraging two types of untrained feature extractors – UGCN and GRNF – the proposed method provides robust and expressive node embeddings, resistant to catastrophic forgetting. Extensive experiments in the OCGL setting demonstrate that when paired with SLDA, this approach significantly outperforms other Continual Learning strategies, including state-of-the-art replay-based methods tailored for graph data. The method achieves performance close to joint offline training across various benchmarks. Beyond strong performance, its efficient streaming updates and no reliance on memory buffers make it a scalable and practical approach to deal with real-time classification on graph node streams.

**Limitations and future directions** While we provide motivation and empirical evidence of the improved resistance to forgetting and higher prediction accuracy of our approach, a thorough theoretical analysis is yet to be developed. Secondly, we highlight how our results are specific for the challenging OCGL scenario, while for different, offline settings other methods can perform more favorably. Finally, we focus on node-level classification, as graph-level is less interesting for OCGL, and leave regression and edge-level tasks as future research.

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

## A    BENCHMARKS

The benchmarks for out experiments are obtained from six node-level classification graph datasets. The CoraFull dataset Bojchevski & Günnemann (2018) is a citation network where nodes represent research papers and edges denote citation links between them, with labels corresponding to paper topics. Amazon Computer Shchur et al. (2019) is a co-purchase graph, with nodes representing products and edges indicating frequent co-purchases in the computer category on Amazon. Arxiv Hu et al. (2021) is a larger citation network based on arXiv submissions in the Computer Science domain. The Reddit dataset Hamilton et al. (2017) comprises posts from various Reddit communities, where each node represents a post, and edges connect posts that were commented on by the same user, capturing user interaction patterns. Roman Empire Platonov et al. (2022) is an heterophilous dataset constructed from the corresponding Wikipedia page, where nodes are words linked through syntactic relationships or adjacency in the text. Lastly, the Elliptic dataset Weber et al. (2019) is a graph of Bitcoin transactions, with edges representing the flow of funds. Only a subset of nodes are labeled as either *licit* (42,019 nodes) or *illicit* (4,545 nodes) transactions. Summary statistics for the six datasets are provided in Table 4.

| Dataset | CoraFull | Amazon Computer | Arxiv | Reddit | Roman Empire | Elliptic |
|---------|----------|-----------------|-------|--------|--------------|----------|
| # nodes | 19,793 | 13,752 | 169,343 | 227,853 | 22,662 | 203,769 |
| # edges | 130,622 | 491,722 | 1,166,243 | 114,615,892 | 32,927 | 234,355 |
| # classes | 70 | 10 | 40 | 40 | 18 | 2 |

Table 4: Dataset statistics.

## B    METRICS

Due to the way the node stream is built, with a definition of task boundaries, we can make use of two commonly adopted continual learning (CL) metrics: *Average Performance (AP)* and *Average Forgetting (AF)* Lopez-Paz & Ranzato (2017). These metrics are both derived from the more general performance matrix $M \in \mathbb{R}^{T \times T}$, where $T$ denotes the total number of tasks, and each element $M_{i,j}$ corresponds to the test performance on task $j$ after training on task $i$.

The *Average Performance* is given by AP $= \frac{1}{T} \sum_{i=1}^{T} M_{T,i}$, representing the model's performance on all tasks after completing the full training stream. The *Average Forgetting* is computed as AF $= \frac{1}{T-1} \sum_{i=1}^{T-1} M_{T,i} - M_{i,i}$, and quantifies how much the model's performance on each task has deteriorated between its initial learning and the end of training. For evaluating performance, we rely on classification accuracy across all datasets, except for Elliptic, which is significantly imbalanced. For this dataset, we instead report the F1 score specific to the *illicit* class.

To track model behavior throughout the node stream, we also employ anytime evaluation: the model is evaluated on validation nodes after each mini-batch update Koh et al. (2021). This provides a fine-grained view of model performance over time, revealing its adaptability to distributional shifts. We quantify this using the *Average Anytime Performance (AAP)* metric Caccia et al. (2021), a generalization of average incremental accuracy for the online scenario. Letting AP$t$ denote the average accuracy after processing the $t$-th mini-batch, and $n$ be the total number of mini-batches, AAP is defined as AAP $= \frac{1}{n} \sum t = 1^n \mathrm{AP}_t$. This metric can be interpreted as the area under the accuracy curve across the training process Koh et al. (2021).

## C    ONLINE FEATURE CENTERING TRICK

As in our experiments we consider also using a standard linear layer trained continually with gradient descent instead of SLDA, in this case it is beneficial to have featured centered in the origin. This is especially true due to the online setting, as having centered features can make the learning of bias parameters for newly observed classes faster, since we initialize the weights symmetrically around zero. Therefore, we adopt an online centering procedure, which allows us to keep the features centered at any point during the stream. Specifically, we maintain an updated global cumulative

mean $m$ of node embeddings, so that

$$m^{(t)} = \frac{(t-1)m^{(t-1)} + \boldsymbol{z}^{(t)}}{t} \, , \tag{11}$$

where for ease of notation we use $\boldsymbol{z}^{(t)} = \boldsymbol{z}_{v_t}^{(t)}$. With this, we then feed the centered embeddings $\boldsymbol{z}_v^{(t)} - m^{(t)}$ in equation (3) of the paper. To ensure consistency of model predictions with online feature centering under updates of the embedding mean as in equation 11, we want to correct the bias term $\boldsymbol{b} \to \boldsymbol{b}'$ to ensure that for any $\boldsymbol{z} \in \mathbb{R}^d$ predictions don't change, that is

$$\boldsymbol{W}(\boldsymbol{z} - m^{(t)}) + \boldsymbol{b}' = \boldsymbol{W}(\boldsymbol{z} - m^{(t-1)}) + \boldsymbol{b} \, . \tag{12}$$

Simplifying and using the definition of the update of $m^{(t)}$ in equation 11, we obtain:

$$-\boldsymbol{W}m^{(t)} + \boldsymbol{b}' = -\boldsymbol{W}m^{(t-1)} + \boldsymbol{b} \, , \tag{13}$$

$$-\frac{(t-1)}{t}\boldsymbol{W}m^{(t-1)} - \frac{1}{t}\boldsymbol{W}\boldsymbol{z}^{(t)} + \boldsymbol{b}' = -\boldsymbol{W}m^{(t-1)} + \boldsymbol{b} \, , \tag{14}$$

$$\boldsymbol{b}' = -\frac{1}{t}\boldsymbol{W}m^{(t-1)} + \frac{1}{t}\boldsymbol{W}\boldsymbol{z}^{(t)} + \boldsymbol{b} \, , \tag{15}$$

$$\boldsymbol{b}' = \frac{1}{t}\boldsymbol{W}(\boldsymbol{z}^{(t)} - m^{(t-1)}) + \boldsymbol{b} \, . \tag{16}$$

This is the bias update formula, and a similar one can be derived for mini-batch updates. We highlight how this bias correction is performed only for seen classes, as we grow the classification head when new classes are encountered. Also, this online centering procedure is not performed with SLDA, since it is a prototype-based classifier and is thus indifferent to feature centering. Empirically, we have observed that this trick greatly improves performance when the node embeddings are not centered, which is often the case with GRNF.

## D    HYPERPARAMETERS

We perform model selection using a limited section of the node stream, approximately 20% of the tasks. Therefore, for class-incremental stream we validate over 7 out 35 tasks for CoraFull, 2 out of 5 for Amazon Computer (as considering 20% of the tasks would mean using only 1, thus without any CL aspect), 4 out of 20 for Arxiv and Reddit, and 2 out of 9 for Roman Empire. For the time-incremental stream, we validate over the first 20% of the nodes (i.e., 2 out of 10 tasks). The hyperparameters are selected by running a standard grid search, over the search space that we illustrate here. For all experiments and both backbones, we consider the gain hyperparameter for weight initialization in $\{0.1, 1, 10\}$. For all methods, except SLDA, we select the learning rate from the set $\{0.01, 0.001, 0.0001, 0.00001\}$, and the number of passes on each batch before going to the next one between 1 and 5. For ER and A-GEM, we consider the proportion of memories to use with respect to each training batch in $\{1, 2, 3\}$. Additionally, we set the same memory buffer size as Donghi et al. (2025): 4000 for CoraFull, Amazon Computer, Roman Empire and Elliptic, 16000 for Arxiv and Reddit. The regularization hyperparameter for EWC and MAS is selected in $\{10^0, 10^2, 10^4, 10^6, 10^8, 10^{10}\}$. For LwF, we consider lambda_dist in $\{1, 10\}$, T in $\{0.2, 2\}$ and the number of mini-batches after which to update the teacher model in $\{10, 100\}$.

We highlight how the only hyperparameter considered for SLDA is the gain of the backbone, making it even easier to use than other methods, avoiding expensive hyperparameter search.

## E    RESULTS WITH TRAINED GCN

In Table 5 we report the AP of CL strategies when used with a trained GCN in the OCGL setting Donghi et al. (2025). These results are obtained in the same configurations used for the experiments in the main paper, only with a trained 2-layer GCN with 256 hidden units instead of an untrained feature extractor. The results on Arxiv are provided only for the class-incremental stream. For the full results with the other metrics (AAP and AF), we point the reader to the original paper introducing the setting.

| METHOD | CORAFULL | A. COMPUTER | ARXIV | REDDIT | ROMAN E. | ELLIPTIC |
|---|---|---|---|---|---|---|
| EWC | $28.10_{\pm 2.76}$ | $14.86_{\pm 6.00}$ | $4.81_{\pm 0.08}$ | $4.33_{\pm 2.77}$ | $8.85_{\pm 0.05}$ | $51.08_{\pm 1.10}$ |
| LwF | $15.74_{\pm 1.56}$ | $19.33_{\pm 0.14}$ | $4.79_{\pm 0.08}$ | $13.13_{\pm 1.92}$ | $8.81_{\pm 0.01}$ | $50.79_{\pm 1.36}$ |
| MAS | $8.40_{\pm 0.62}$ | $12.68_{\pm 8.28}$ | $3.35_{\pm 0.99}$ | $10.21_{\pm 1.03}$ | $11.02_{\pm 1.36}$ | $51.08_{\pm 1.10}$ |
| TWP | $13.98_{\pm 1.66}$ | $19.80_{\pm 3.41}$ | $4.74_{\pm 0.05}$ | $12.79_{\pm 2.51}$ | $8.99_{\pm 0.06}$ | $\mathbf{51.13_{\pm 1.74}}$ |
| ER | $20.65_{\pm 2.33}$ | $38.53_{\pm 2.93}$ | $23.43_{\pm 1.65}$ | $22.34_{\pm 2.46}$ | $10.43_{\pm 0.20}$ | $43.94_{\pm 0.52}$ |
| A-GEM | $\mathbf{40.45_{\pm 0.77}}$ | $38.49_{\pm 2.80}$ | $17.16_{\pm 1.45}$ | $\mathbf{58.08_{\pm 8.04}}$ | $9.07_{\pm 0.15}$ | $47.06_{\pm 1.18}$ |
| PDGNN | $38.48_{\pm 1.15}$ | $68.91_{\pm 0.33}$ | $\mathbf{35.86_{\pm 1.20}}$ | $53.98_{\pm 0.44}$ | $\mathbf{14.20_{\pm 0.87}}$ | $49.91_{\pm 1.44}$ |
| SSM-ER | $23.23_{\pm 2.69}$ | $\mathbf{70.45_{\pm 3.66}}$ | $31.28_{\pm 1.91}$ | $50.48_{\pm 1.69}$ | $11.71_{\pm 0.51}$ | $24.45_{\pm 2.97}$ |
| SSM-A-GEM | $36.63_{\pm 4.63}$ | $50.01_{\pm 9.12}$ | $13.12_{\pm 2.46}$ | $22.54_{\pm 4.51}$ | $9.00_{\pm 0.09}$ | $40.48_{\pm 0.82}$ |
| BARE | $12.59_{\pm 0.82}$ | $20.51_{\pm 4.26}$ | $4.74_{\pm 0.08}$ | $12.59_{\pm 2.59}$ | $8.78_{\pm 0.10}$ | $51.28_{\pm 2.37}$ |
| JOINT | $67.55_{\pm 0.05}$ | $83.07_{\pm 1.30}$ | $58.58_{\pm 0.28}$ | $90.02_{\pm 0.12}$ | $39.47_{\pm 0.33}$ | $71.97_{\pm 0.83}$ |

Table 5: AP results in the OCGL setting from (Donghi et al., 2025) of CL strategies with trained GCN and joint offline training upper bound. In particular, TWP (Liu et al., 2021), PDGNN (Zhang et al., 2024a), SSM-ER and SSM-A-GEM (Zhang et al., 2022b) are graph-specific CL baselines. Best performing method for each dataset is highlighted in bold. For Arxiv class-incremental stream is considered.

## F  PERFORMANCE PLOTS

We report here plots that show model performance along the node stream, to provide a more detailed understanding of the dynamics of training and forgetting. In Figures 2 and 3 we plot for each benchmark a comparison of the performance using the considered methods, for UGCN and GRNF backbones respectively. We highlight the task boundaries with dotted vertical lines, with the thicker dashed one indicating the threshold at which hyperparameter selection is performed. The upper bound of joint training on data up to the present task is represented as an horizontal line over the batches of each task.

In Figures 4-17, we instead illustrate a more detailed breakdown of model performance: for each benchmark, backbone and considered method, we plot the results of anytime evaluation broken down on individual tasks, allowing a better understanding of when and where forgetting occours.

## G  LLM USAGE

Large Language Models were used for the purpose of text editing.

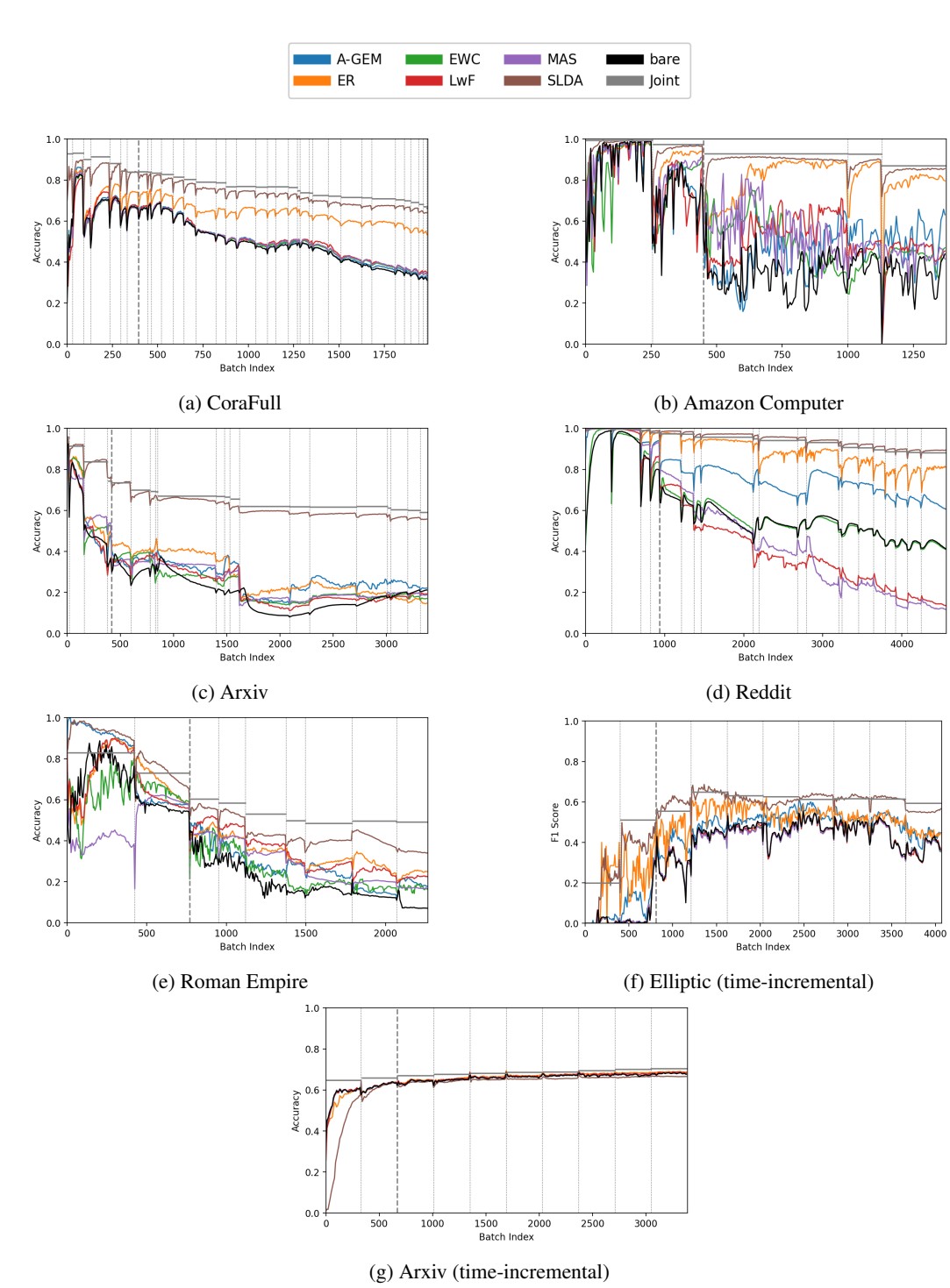

Figure 2: Anytime evaluation performance for the different datasets with UGCN Backbone. We highlight with vertical lines the task boundaries and the hyperparameter selection threshold.

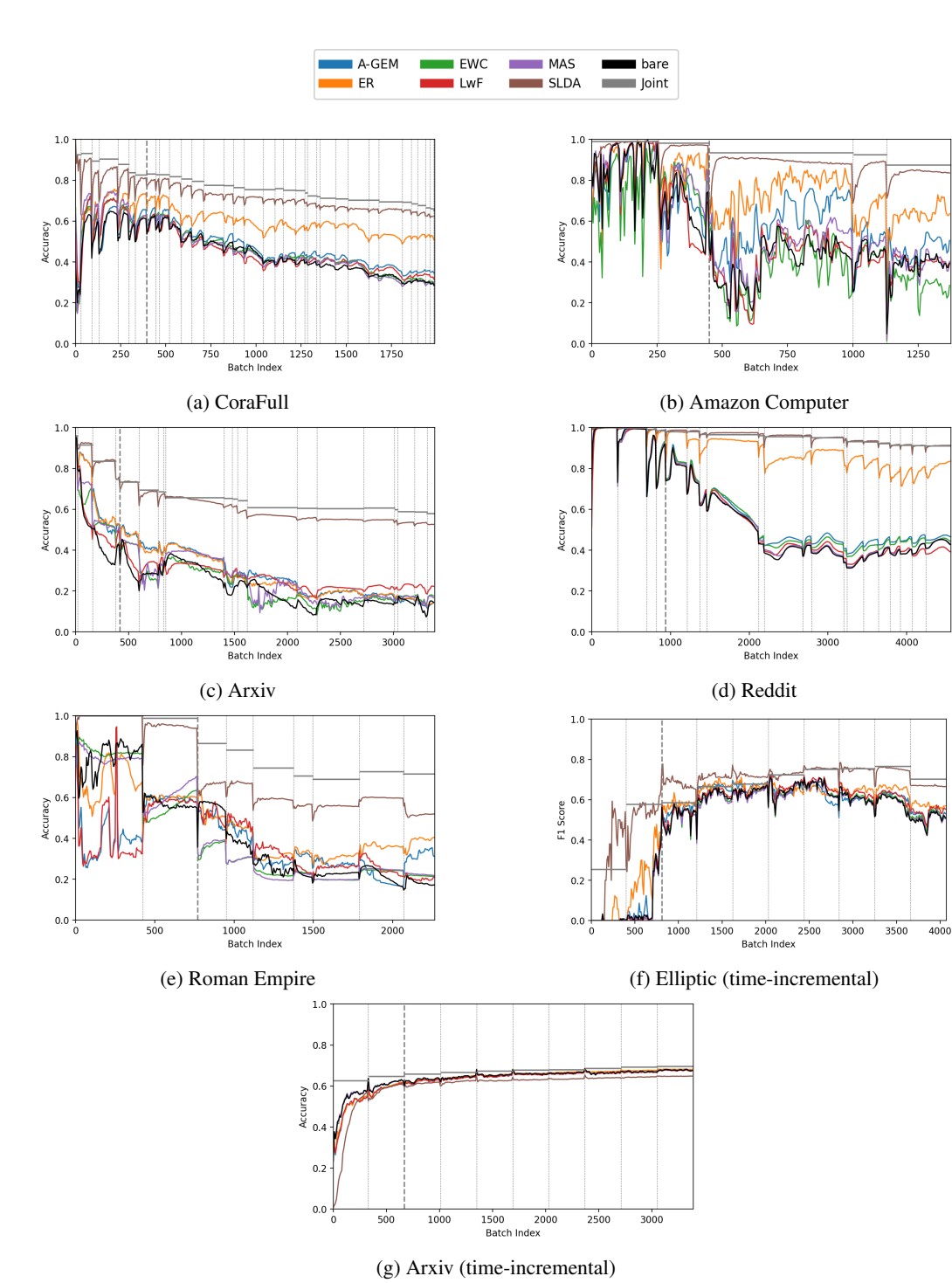

Figure 3: Anytime evaluation performance for the different datasets with GRNF backbone. We highlight with vertical lines the task boundaries and the hyperparameter selection threshold.

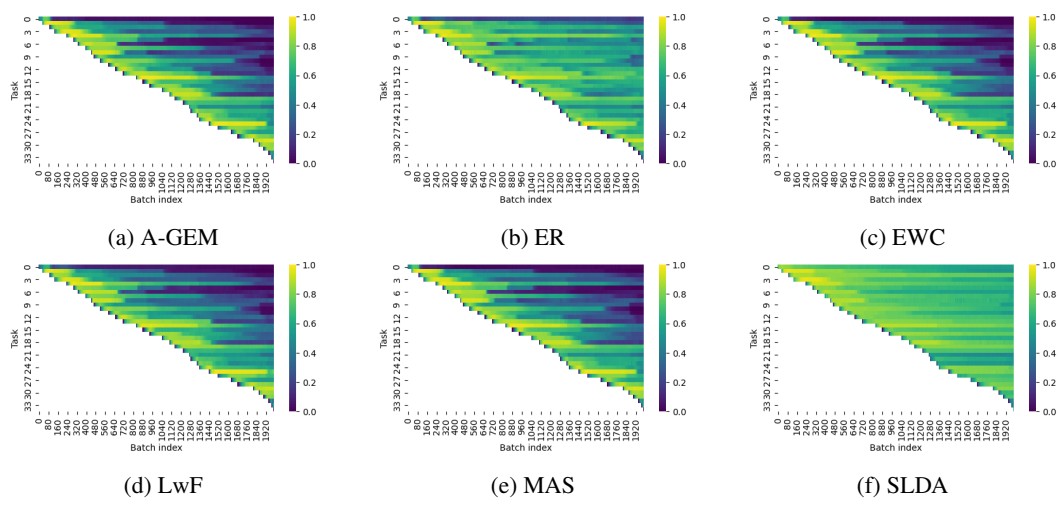

(a) A-GEM       (b) ER       (c) EWC

(d) LwF       (e) MAS       (f) SLDA

Figure 4: Anytime evaluation by task for the CoraFull dataset with UGCN backbone.

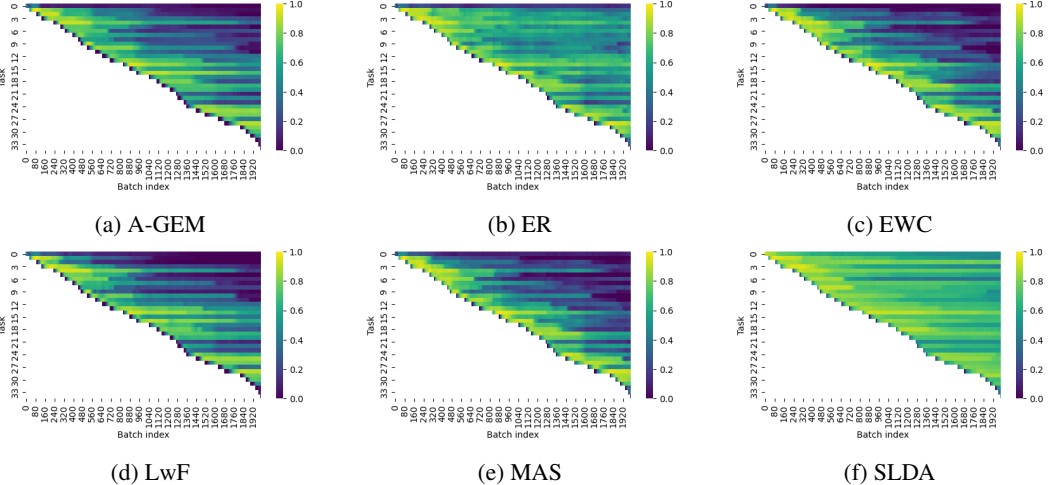

(a) A-GEM       (b) ER       (c) EWC

(d) LwF       (e) MAS       (f) SLDA

Figure 5: Anytime evaluation by task for the CoraFull dataset with GRNF backbone.

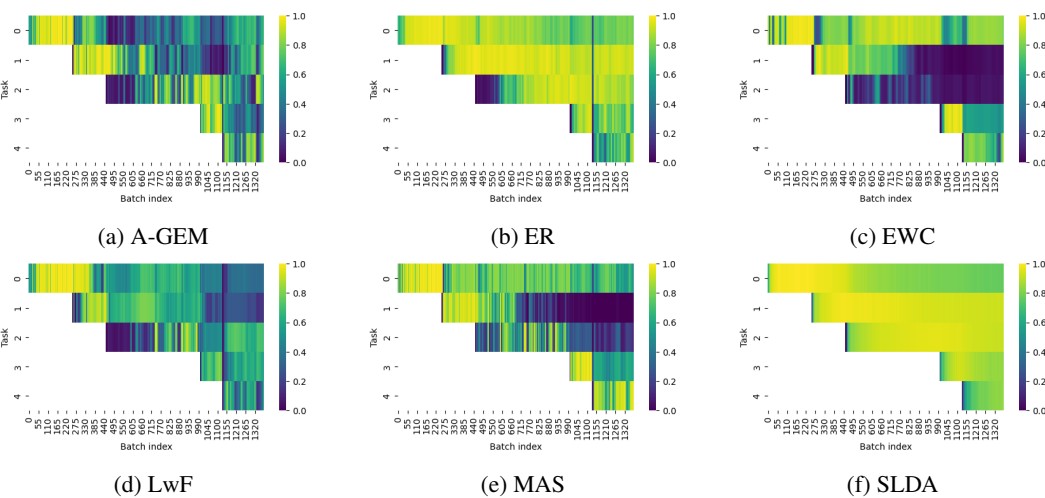

(a) A-GEM       (b) ER       (c) EWC

(d) LwF       (e) MAS       (f) SLDA

Figure 6: Anytime evaluation by task for the Amazon Computer dataset with UGCN backbone.

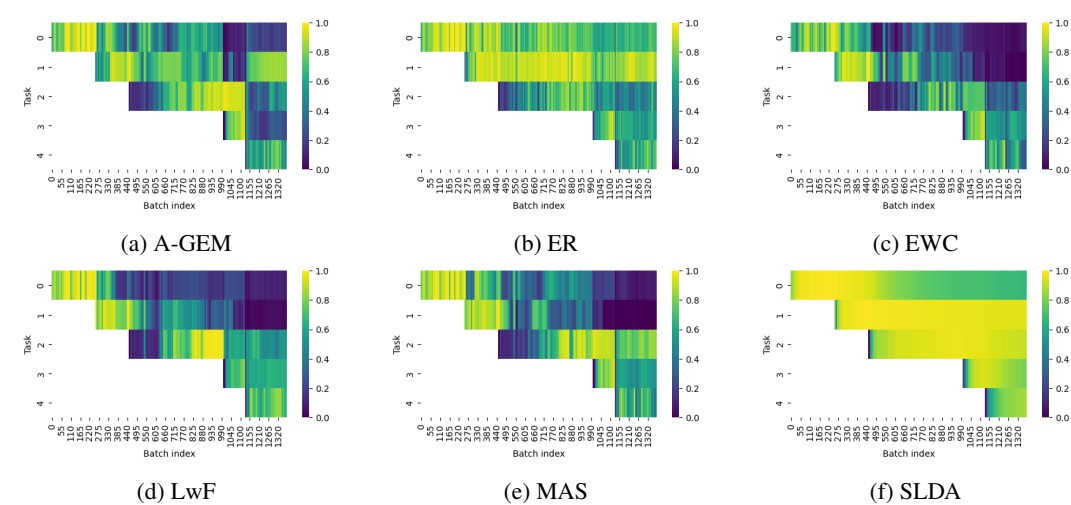

Figure 7: Anytime evaluation by task for the Amazon Computer dataset with GRNF backbone.

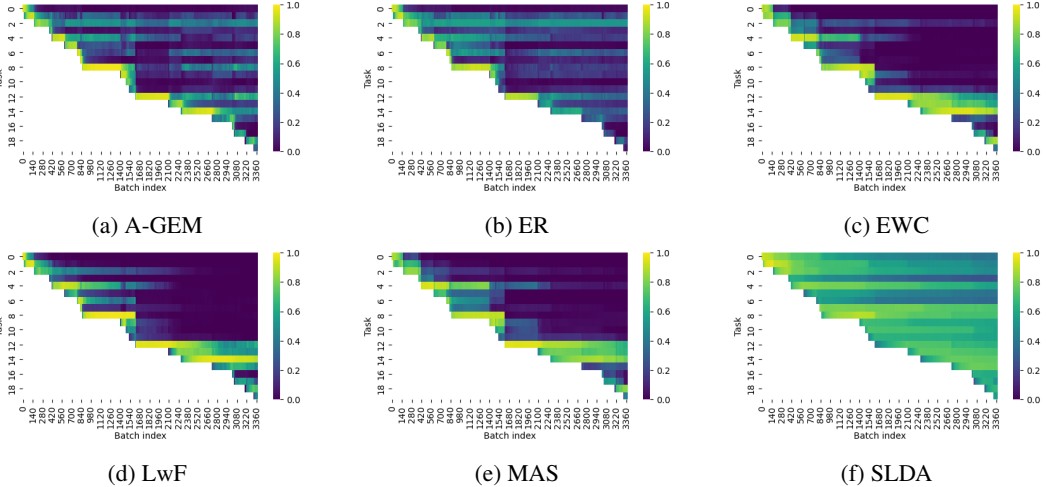

Figure 8: Anytime evaluation by task for the Arxiv dataset with UGCN backbone.

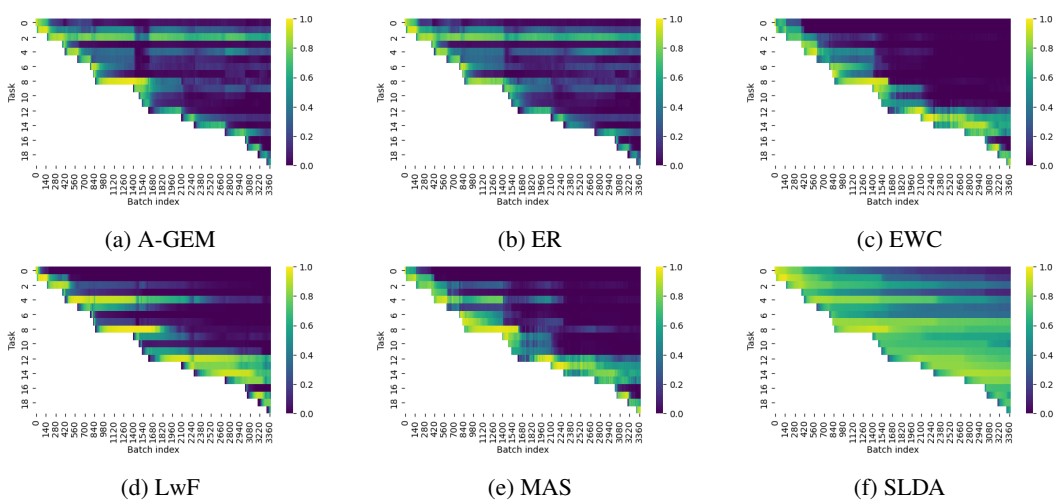

Figure 9: Anytime evaluation by task for the Arxiv dataset with GRNF backbone.

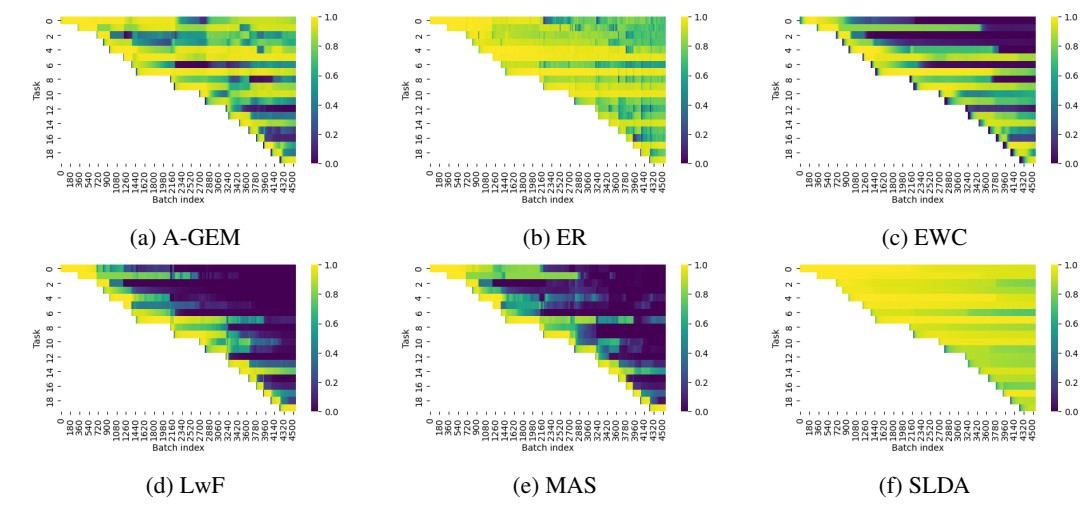

Figure 10: Anytime evaluation by task for the Reddit dataset with UGCN backbone.

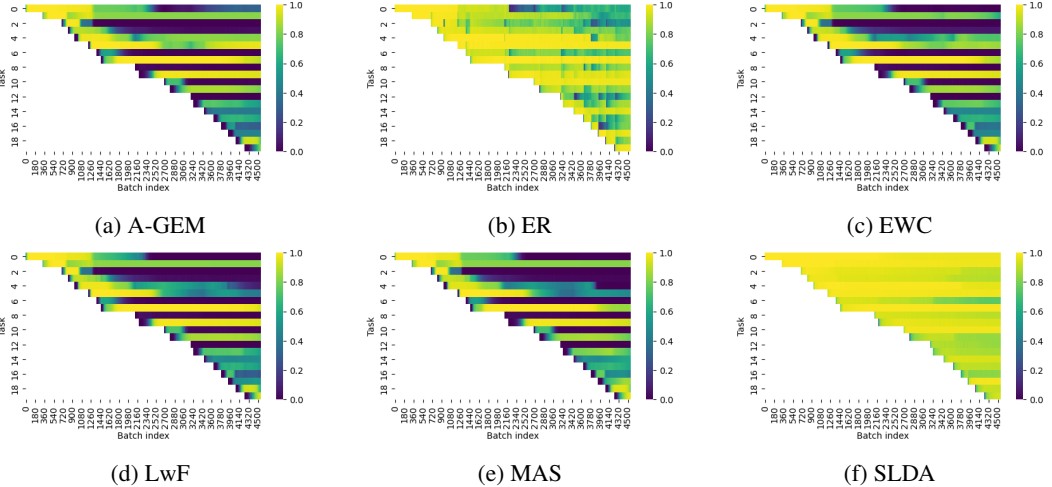

Figure 11: Anytime evaluation by task for the Reddit dataset with GRNF backbone.

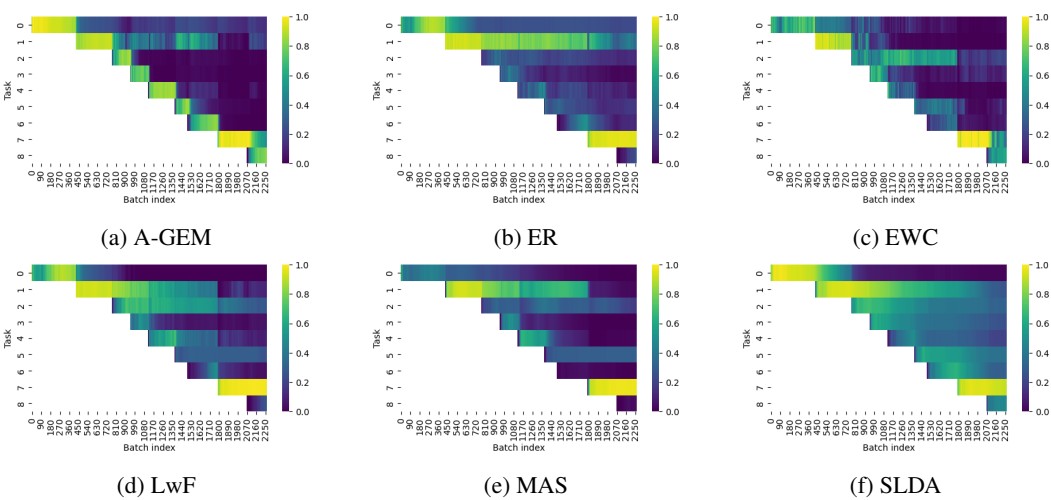

Figure 12: Anytime evaluation by task for the Roman Empire dataset with UGCN backbone.

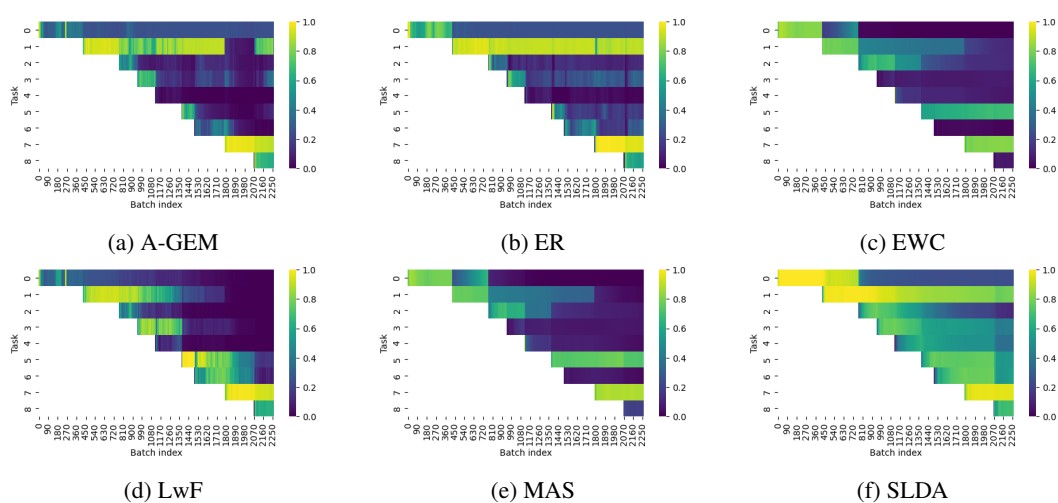

(a) A-GEM

(b) ER

(c) EWC

(d) LwF

(e) MAS

(f) SLDA

Figure 13: Anytime evaluation by task for the Roman Empire dataset with GRNF backbone.

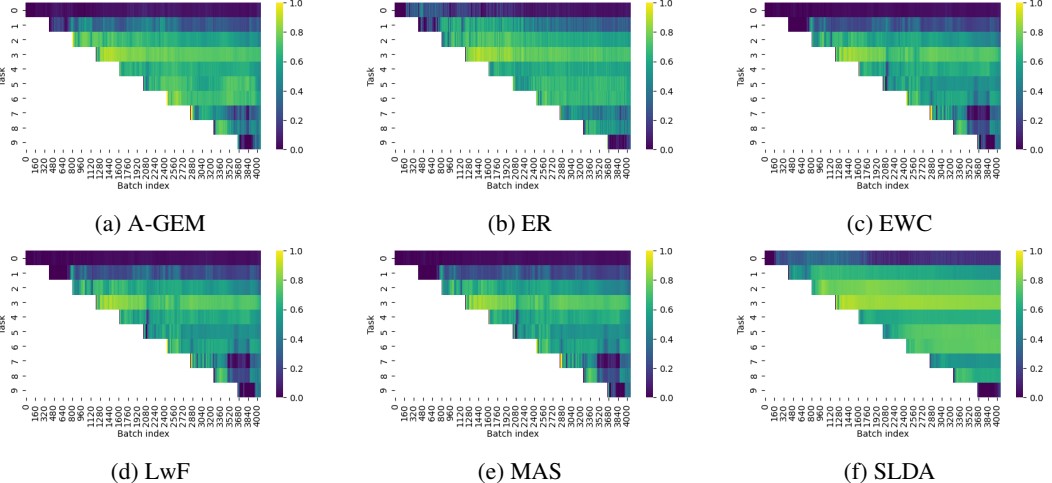

(a) A-GEM

(b) ER

(c) EWC

(d) LwF

(e) MAS

(f) SLDA

Figure 14: Anytime evaluation by task for the Elliptic dataset with UGCN backbone.

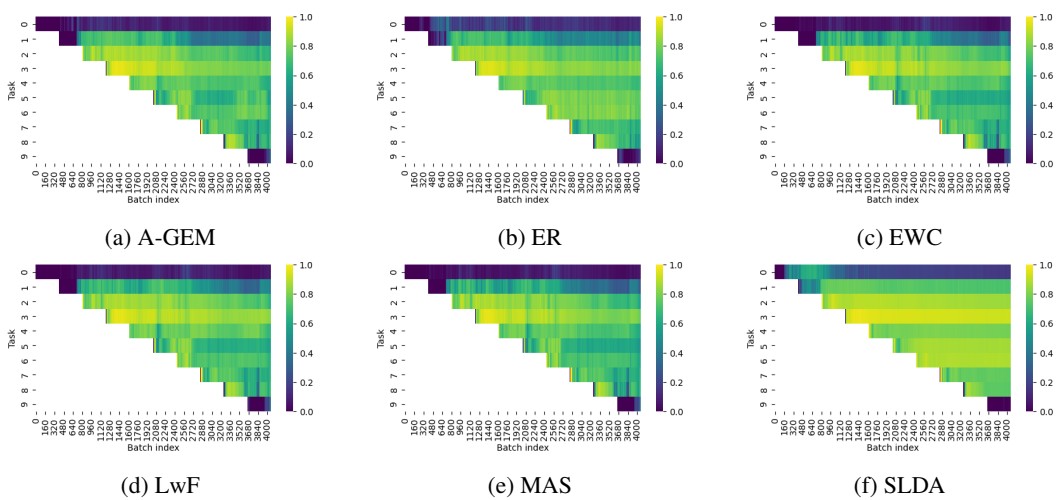

(a) A-GEM

(b) ER

(c) EWC

(d) LwF

(e) MAS

(f) SLDA

Figure 15: Anytime evaluation by task for the Elliptic dataset with GRNF backbone.

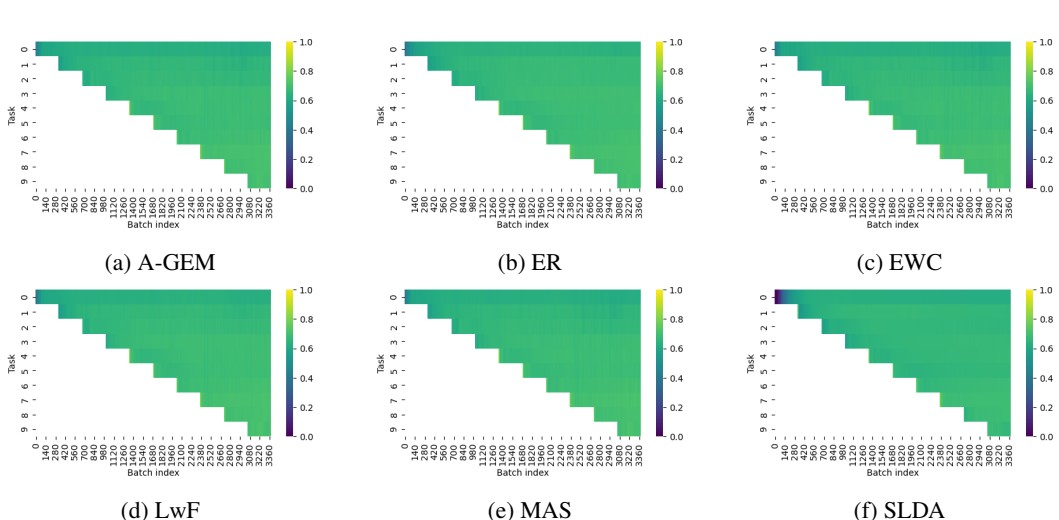

Figure 16: Anytime evaluation by task for the Arxiv dataset with UGCN backbone with time-incremental stream. We note how the very homogeneous performance compared to other benchmarks suggests the absence of significant distribution drifts between tasks.

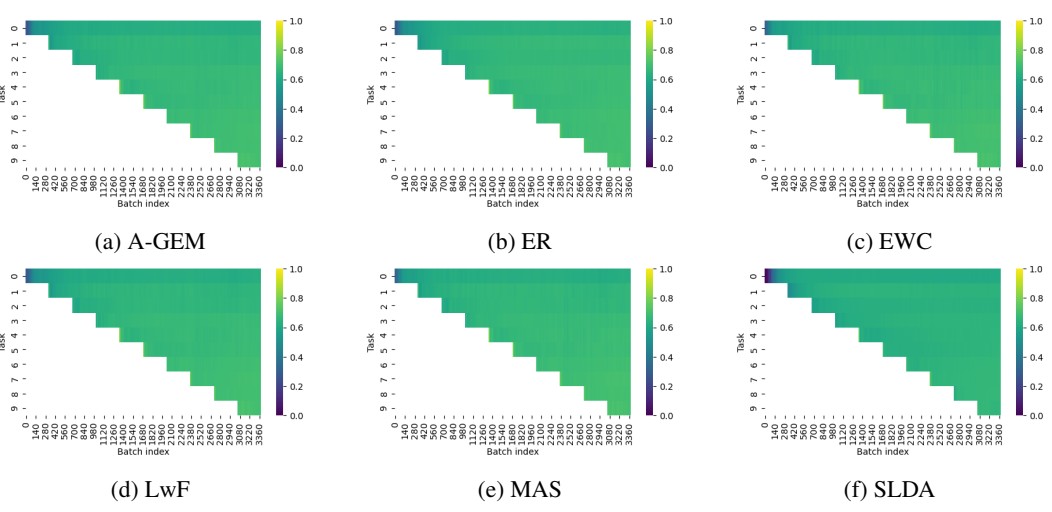

Figure 17: Anytime evaluation by task for the Arxiv dataset with GRNF backbone with time-incremental stream. We note how the very homogeneous performance compared to other benchmarks suggests the absence of significant distribution drifts between tasks.

