# OpenReview forum: "The Unreasonable Effectiveness of Randomized Representations in Online Continual Graph Learning"
_ICLR.cc/2026/Conference — ICLR 2026 Conference Withdrawn Submission_

### Official Review · Reviewer_Nni1 · 2025-10-30

**Soundness:** 2
**Presentation:** 3
**Contribution:** 2
**Rating:** 4
**Confidence:** 3

**Summary:**

This paper investigates the surprising effectiveness of randomized representations in Online Continual Graph Learning (OCGL). The authors propose to replace the learned graph encoder with a fixed random projection encoder, and only train a lightweight streaming linear classifier (SLDA) during continual learning. This design aims to eliminate representation drift—the major source of catastrophic forgetting—while preserving expressive power through high-dimensional random embeddings. Extensive experiments on several graph continual learning benchmarks show that such a simple approach not only matches but often surpasses state-of-the-art methods, and even approaches the jointly trained oracle performance.

**Strengths:**

1. The paper is clearly written and easy to follow, with clear motivation and logical presentation.
2. Extending the concept of random representations from image-based continual learning to graph case is sound.
3. The method is elegant and simple—freezing the encoder effectively prevents forgetting while keeping the system lightweight and stable.

**Weaknesses:**

1. The novelty is limited. As the method mainly transfers the existing idea from [1] to the graph domain.
2. The paper lacks theoretical analysis on how well the random graph embeddings can approximate the oracle representations or under what conditions they remain expressive.
3. Fixing the encoder reduces model plasticity, limiting adaptability to evolving or dynamic graph structures.
4. The baselines used for comparison are relatively old, making it difficult to fully assess performance against the latest continual learning models.

**Questions:**

Compared to the approach in [1], which studies the effectiveness of random representations in general online continual learning, what are the unique challenges or characteristics of the graph continual learning scenario that make this problem distinct?

[1] Prabhu A., Sinha S., Kumaraguru P., et al. Random Representations Outperform Online Continually Learned Representations. arXiv preprint arXiv:2402.08823, 2024.

---

> ### Author Response · Authors · 2025-11-21
>
> - **W1, Q1**: Paper [1] is indeed a related one, which we mention in our related work section as the NeurIPS paper titled "RanDumb: Random Representations Outperform Online Continually Learned Representations". Our research began before [1] was available and was developed independently. While both studies explore random representations paired with a lightweight classifier, the contributions diverge in significant ways. Our work addresses the OCGL setting, which involves challenges that do not arise in the standard OCL scenario for images considered in [1]. Working on a node stream within an evolving graph introduces the issue of structural drift, as node neighborhood distribution may change through time, even for the same node, potentially negating the benefits of the randomized approach. Nonetheless, we show empirically that UGCN and GRNF are robust against the considered shift. Additionally, unlike in the image domain, graph continual learning lacks broadly applicable pretrained backbones. Our approach therefore serves as a domain-appropriate alternative: a frozen, untrained encoder that requires no memory buffer and yet achieves accuracy close to joint offline training on a wide range of OCGL benchmarks.
>     Thus, our paper addresses a graph-specific gap in the literature. These differences reflect distinct problem settings, domain assumptions, and findings, resulting in contributions that are complementary rather than overlapping.
> - **W2**: For a theoretical analysis of the expressive power of our randomized feature extractors, we refer to the original papers and others from the literature on randomized models (referenced in our manuscript). In particular, GRNF ones are thoroughly studied in the papers by Maron et al. (2019), Keriven \& Peyre (2019), and Zambon et al. (2020), showing universal approximation capabilities. Provided a sufficient number of features is considered (most of the available guarantees are asymptotic), they remain expressive even under the class-incremental setting as they are not tailored on any training data from the stream. With new classes added to the problem, performance may degrade due to the increased complexity of classifying between more classes, but this is true also for trained models, and more units may be required to maintain the same performance. We will add a comment on this point. Thank you.
> - **W3**: Fixing the encoder is a strength of our approach, as we show that it is enough to adapt the classifier to achieve superior performance, provided that expressive features are considered. Our results with SLDA demonstrate that this allows us to obtain remarkably high performance, almost on par with joint offline training (an upper bound on what CL methods can achieve).
> Thus, we argue that fixing the encoder allows us to reach an excellent tradeoff between stability and plasticity, rather than limiting adaptability.
> - **W4**: Please note that Appendix E considers and discusses recent state-of-the-art CL methods for graphs, with best results summarized in Table 3 of the main paper, highlighting performance improvements of untrained feature extraction coupled with SLDA; we understand that they might be somewhat hidden, and we will reference them better in the main paper. On the other hand, Tables 1 and 2 in the main paper report results for the most widely used CL strategies, which remain standard baselines in the literature despite being older.

---

### Official Review · Reviewer_yf4i · 2025-10-30

**Soundness:** 2
**Presentation:** 2
**Contribution:** 2
**Rating:** 2
**Confidence:** 4

**Summary:**

This paper studies Online Continual Graph Learning (OCGL), where nodes arrive sequentially and the model must adapt to evolving distributions without access to past data. The authors combine a frozen randomized GNN encoder with a lightweight online classifier, claiming improved stability and reduced forgetting. Experiments on several node-classification graph datasets report competitive performance.

**Strengths:**

1. Motivates the challenge of catastrophic forgetting when nodes arrive sequentially and past data cannot be stored.

2. Uses a simple and intuitive strategy: freezing the encoder to avoid representation drift.

3. Empirical evaluation across multiple node-classification graph datasets, including some real-world application datasets.

**Weaknesses:**

1. Problem definition lacks clarity. The introduction states that *“nodes arrive sequentially and undergo distribution drifts,”* and *"an incremental graph G, induced by a stream of nodes v1, v2, . . . , vt, . . . that are added one by one,"* but this scenario also exists in the temporal graph continual learning setting. It is unclear what concrete challenge OCGL uniquely poses beyond existing temporal GNN continual learning methods.  Authors must clarify why temporal graph continual learning baselines are excluded and whether those models already address the studied challenge.

2. Motivation for real-time online learning is underdeveloped. The paper claims offline training is infeasible, yet it does not provide practical evidence or runtime constraints where online learning is strictly required. A discussion contrasting the cost-benefit vs. offline retraining is needed.

3. Limited technical novelty. The core method is freezing a randomized encoder and training a streaming classifier. Each component has been previously studied (random features, SLDA, UGCN, GRNF). The contribution appears to be an empirical combination rather than a principled algorithmic innovation.

4. Method description is ambiguous in Table 1. Table 1 lists baselines but does not clearly show where the proposed method sits in the taxonomy. This table is very confusing, it does not even highlight the proposed method or mark the best-performing results.

5. Baseline coverage is insufficient. There are several continual graph learning baselines (e.g., ER-GNN [1], TWP [2]) and temporal graph continual learning (N-ForGOT [3], OTGNet [4]) works that should be compared. Current baselines do not convince that the method advances the state of the art.



[1] Fan Zhou and Chengtai Cao. Overcoming catastrophic forgetting in graph neural networks with experience replay. In Thirty-Fifth AAAI Conference on Artificial Intelligence, AAAI 2021, Thirty-Third Conference on Innovative Applications of Artificial Intelligence, IAAI, pp. 4714–4722, 2021.

[2] Huihui Liu, Yiding Yang, and Xinchao Wang. Overcoming catastrophic forgetting in graph neural networks. In Thirty-Fifth AAAI Conference on Artificial Intelligence, pp. 8653–8661. AAAI Press, 2021.

[3] Wang, Liping, et al. "N-ForGOT: Towards Not-forgetting and Generalization of Open Temporal Graph Learning." The Thirteenth International Conference on Learning Representations, ICLR, 2025.

[4] Kaituo Feng, Changsheng Li, Xiaolu Zhang, and Jun Zhou. Towards open temporal graph neural networks. In The Eleventh International Conference on Learning Representations, ICLR, 2023.

**Questions:**

See Weaknesses

---

> ### Author Response · Authors · 2025-11-21
>
> - **W1**: While the OCGL and Open Temporal Graph Learning (OTGL) settings share the similarity of operating on a growing and evolving graph, they are fundamentally distinct both in terms of the graph stream construction and of the challenges they face.
>     OCGL considers a stream of nodes with associated neighbors for which the model needs to make predictions. Each of these nodes is observed only once (either alone or within a very small batch), used to update the model, and then discarded.
>     An OCGL model needs to promptly update while adhering to tight computation and memory budgets, even as the underlying graph grows.
>     On the other hand, OTGL is defined over a stream of temporal links with associated modeling challenges. While in principle this is compatible with an online setting, the works we are aware of (as those mentioned by the reviewer) all consider snapshots of temporal graphs, which are incompatible with online learning on graphs.
>     As such, we think that OTGL results closer to offline Continual Graph Learning (CGL) rather than to an online setting.
>     We thank the reviewer for pointing out the similarity of our setting to OTGL, and we will make sure to refer to the relevant papers [3] and [4] in the related works.
> - **W2**: Online learning, and therefore also Online CL and OCGL, is motivated by the need to operate under tight memory and computation budgets while updating promptly and providing anytime predictions. Our paper does not claim that offline retraining is infeasible in absolute terms. Rather, it focuses on scenarios where repeatedly retraining a model on all data is impractical due to these constraints. In such contexts, maintaining a model that can update efficiently and continuously from single (or small batches of) data points is often preferable, or even required, depending on the application; e.g., when dealing with edge devices or in privacy-sensitive applications. The computational and memory constraints entail a tradeoff in prediction accuracy, which can be quantified by comparing the results for the "Joint" baseline with the OCGL methods.
>     These considerations are rather common in the CL/OCL literature; this is why we did not provide an in-depth discussion on the topic.
> - **W3**: Our contribution lies in combining randomized graph encoders with SLDA -- two approaches coming from different research fields -- in a principled way that is specifically tailored to OCGL, as motivated in the paper. This is the first work studying this combination and showing remarkable empirical results, to our knowledge.
>     Specifically, randomized graph feature extractors (UGCN and GRNF) enable the use of SLDA for CL and graphs, otherwise not applicable, resulting in expressive and stable models that achieve high accuracy and are robust to forgetting, consistently outperforming more complex CL methods by a large margin, demonstrating that strong OCGL performance can be achieved without heavy engineering. Moreover, using random features in an online setting with structural drift is nontrivial, as node neighborhoods evolve over time. We provide the first empirical evidence that such features remain robust under these conditions.
>     For these reasons, we believe that ours is a substantial contribution to the community.
> - **W4**: Our approach corresponds to the entire SLDA line of Table 1, consistently achieving better results. Table 1 includes results for three different metrics on multiple datasets, multiple CL methods, and two different backbones, which may have resulted in an unclear presentation. We will make sure to highlight the rows with SLDA that refer to our approach, and also mark the best and second-best result for each dataset-backbone pair.
> - **W5**: The results in the main body of the paper are obtained with popular CL strategies on features extracted with UGCN and GRNF. For a comparison with more recent state-of-the-art CGL methods, we refer to Table 3, which reports the best results obtained with a trained GNN backbone; we also emphasize that these results are reported in full in Appendix E. In particular, we report results for the requested TWP and with ER on a GNN (not ER-GNN precisely as it is not directly compatible with the online setting; we refer to [5], Section IV, for more details). As for the suggested N-ForGOT and OTGNet, we observe that they are defined for OTGL rather than OCGL: specifically, they are incompatible with the online requirements (see the representative triad selection of OTGNet, or the computation of divergence $div(G_{k-1}, G_k)$ in N-ForGOT) and thus unsuitable for the benchmarking of the present paper.
>
> [5] Donghi et al. Online Continual Graph Learning.  arXiv preprint arXiv:2508.03283, 2025.

---

### Official Review · Reviewer_3MG8 · 2025-10-31

**Soundness:** 3
**Presentation:** 3
**Contribution:** 2
**Rating:** 4
**Confidence:** 3

**Summary:**

The paper tackles the problem of Online Continual Graph Learning (OCGL), where nodes arrive sequentially and models must learn from a single pass of data while retaining past knowledge. The authors propose a simple decoupled approach that combines randomized, untrained graph feature extractors (UGCN or GRNF) with a Streaming Linear Discriminant Analysis (SLDA) classifier. This design removes the need for training a GNN backbone and aims to prevent catastrophic forgetting while keeping computation and memory low. Experiments on seven benchmarks show that the method achieves state-of-the-art or near-upper-bound accuracy, outperforming existing OCGL and CL baselines. Overall, the paper argues that fixed randomized representations can provide strong and stable performance for online graph learning.

**Strengths:**

1.  The paper proposes a decoupled OCGL framework using untrained randomized graph features with a lightweight SLDA classifier. The approach is easy to implement yet very effective.

2. The method is computationally lightweight, requires no replay buffer, and scales well to streaming graph data. This makes it practically appealing for real-time or resource-limited continual learning scenarios.

3. The method is tested on seven benchmarks and consistently achieves state-of-the-art or near-upper-bound performance without using replay buffers.

**Weaknesses:**

1. In the proposed method, node embeddings are generated once and never updated. However, as new nodes arrive and the graph structure evolves, the context of old nodes may change, making their fixed embeddings potentially outdated. This raises concerns about the reliability of predictions for earlier nodes. The paper does not provide experiments or analysis to assess this effect.

2. The authors emphasize that the method is lightweight, but there are no quantitative comparisons of runtime, memory usage, or computational complexity against other OCGL approaches.

3. The paper experiments with two random feature extractors (UGCN and GRNF) without explaining why these were chosen or whether other random graph encoders would behave similarly. A deeper analysis of why these random features perform so well would make the contribution more convincing.

**Questions:**

1. The embeddings of old nodes are fixed once generated. When new nodes and edges arrive, the graph structure changes — could this make old embeddings outdated? Have you tested how this affects performance?

2. You claim the method is lightweight and efficient. Can you provide comparisons of training time, memory usage, or complexity with other OCGL methods?

3. Why did you choose only UGCN and GRNF as random feature extractors? Would other randomized graph encoders work similarly?

4. How would the method perform in real dynamic graphs where node attributes or edges change frequently?

---

> ### Author Response · Authors · 2025-11-21
>
> - **W1, Q1**: Complying with the OCGL setting, we do not store the node embeddings once they are generated, except for the memory buffer of methods that use one. When a prediction is requested for a node, the embedding is obtained with up-to-date information: for a given test node, the model is presented with the most recent neighborhood as input, whether or not there is a change in neighborhood distribution caused by the stream (which is called structural drift in the literature). Of course, the structural drift entails that the classification of the same node $v$ might differ over time as the model input -- the neighborhood of $v$ -- may change with time, while it is the class means of feature stored by SLDA that may become outdated, potentially negatively affecting performance. This is however an inherent challenge of OCGL itself, not a shortcoming related to specific methods. We will elaborate on this aspect further in the revised paper, thank you. To quantify the impact of the structural drift, due to how SLDA works, we can compare the joint upper bound and the SLDA performance, as the gap can be attributed to this phenomenon; e.g., for the homophilic dataset CoraFull the gap is small at about 2\%, while for the heterophilic (thus more prone to structural drift in class-incremental setting) Roman Empire it is about 15\%.
> - **W2, Q2**: The main elements to claim that our approach is lightweight are the absence of a memory buffer and rehearsal, the fact that it does not perform backpropagation through backbone layers, and the empirical observation that we achieve better performance even with a smaller model size.
>     While a thorough profiling would be very sensitive to the considered setup (e.g., backbone architectures, model sizes to achieve desired performance, and implementation), we summarize the improvements as follows. Memory is reduced thanks to a smaller model size (compare Table 3 with Figure 1), the absence of memory buffers, and storing additional tensors for gradient computation is not necessary. In parallel, computation is reduced by avoiding gradient computations and updates for backbone parameters, and also avoiding rehearsal strategies; also the inference step benefits from a reduced model size.
>     In the revised paper, we will facilitate comparison by adding other OCGL methods to Figure 1, showing how they position themselves in terms of the number of hidden units and performance.
> - **W3, Q3**: UGCN and GRNF have been chosen as representative random feature extractors: UGCN is the randomized counterpart of possibly the most popular GNN in use in the literature, the GCN, while GRNF comes from a family that has been proven to be maximally expressive. Of course, as mentioned in the paper, other sufficiently expressive randomized graph encoders (e.g., GESN) should also work, bringing their own inductive biases that may be more effective for some datasets and less for others.
> - **Q4**: A setting where both node attributes or edges change frequently is fairly different from the one explored in this paper, and from the ones considered in the literature on CL for graphs more at large. The CL focus on retaining past knowledge may not be ideal in such a highly dynamic setting.
>     Approaches from temporal graph learning are perhaps more appropriate there.

---

### Official Review · Reviewer_re4q · 2025-11-01

**Soundness:** 3
**Presentation:** 3
**Contribution:** 3
**Rating:** 6
**Confidence:** 2

**Summary:**

This paper studied the research problem of online continual graph learning, where nodes sequentially show up in the graph, resulting a continual learning problem where the graph (dataset) keeps changing and drifting. The proposed method is very simple, just a untrained (randomly initialized) feature extractor model, followed by a continuously trained linear classifier. The authors also provided insights on why this simple model works so well on this task. The authors evaluated the proposed method against multiple baselines on a couple benchmarks, and showcased very good performances.

**Strengths:**

1. The proposed method is very simple and easy to implement.
2. This paper is overall clearly written and easy to follow.
3. The authors conducted fairly comprehensive evaluations, with the proposed method showcasing good improvements.

**Weaknesses:**

1. I'm not super familiar with the literature of OCGL, and found it interesting that the whole setup assumes only new nodes showing up along with their edges, but not consider new edges between old nodes. If we consider social platforms with user-user graphs and/or user-video/product/ads graphs, IMO it would make much more sense to consider an edge-based streaming update, where a new node shows up along with the first edge that connects to it. I'd appreciate if the authors can discuss more regarding this difference in the setup, as well as how would it affect the proposed method's effectiveness/efficiency etc.
2. The datasets seem small. I wonder are there larger datasets with timestamps exist that the authors can evaluate with.

**Questions:**

n/a

---

> ### Author Response · Authors · 2025-11-21
>
> - **W1**: The considered setting closely matches what is typically done in CL for graphs for the node-level class-incremental setting.
>     We would like to clarify that we focus on node-level tasks and on what a model sees: an OCGL model will be presented with a stream of nodes -- alongside their neighbors -- for which a prediction is requested.
>     Accordingly, if a new link is created with a previously seen node, it will be considered and processed by the model when predictions for its related nodes are requested.
>     In this sense, handling an evolving graph that results from a stream of edges, also between previously seen nodes, is a natural extension of our setup.
>     More generally, however, adapting the proposed methods (and also other ones) to an edge-level stream, along with the implications for effectiveness and efficiency, would depend on the type of prediction task associated with the stream: if it remains node classification (so, for example, observing the labels of the two nodes connected by the current edge), current methods could be easily adapted, with the main efficiency change being due to longer stream, while effectiveness would likely slightly increase due to the additional updates (especially for non SLDA methods). A similar setup could be considered for edge classification with similar considerations. For link prediction, on the other hand, more care would be required regarding the availability of data. Overall, we believe that keeping an untrained randomized feature extractor would still be advantageous to prevent forgetting while ensuring enough expressive power to adapt to the stream.
>
> - **W2**: For scale of the datasets, we are comparable with the rest of the literature of CL on graphs. We did not find good, large and publicly available datasets with nodes equipped with timestamps, except for Elliptic and Arxiv (although even Arxiv is not the best suited for our purposes of CL as the drift through time is not really significant). Of course, we remain open to considering other datasets when appropriate.

---

### Note · Authors · 2026-01-22

I have read and agree with the venue's withdrawal policy on behalf of myself and my co-authors.